# Connecting chemical structure to single cell signaling profiles

Hannah L. Thirman [1,2,3], Madeline J. Grider-Hayes [3], Laura C. Geben [1], Rebecca A. Ihrie [1,3], Lauren E. Brown [4], John A. Porco Jr [4] & Jonathan M. Irish [1,2,3] ✉

A challenge in chemical biology is to study structure-activity relationships (SAR) in vivo in cells. Multiplexed activity profiling (MAP), developed for natural product discovery, is well-suited to address this challenge as it is high throughput, singlecell, and measures multiple hallmark cellular functions. Applying MAP while systematically varying molecular structure (SAR-MAP) could reveal previously unappreciated activity within chemical families and prioritize candidate molecules for further characterization. Here we use SAR-MAP to identify structural features responsible for specific bioactivities of the natural product family, rocaglates. MV411 leukemia cells and healthy human leukocytes are selected for proof-of-concept screening. Testing 600 representative molecules using MAP classifies roughly half of tested rocaglates (9 of 19) as bioactive. SAR-MAP elucidates a methoxy substituent on select rocaglate pyrimidinones as responsible for a desirable anti-leukemia activity. Thus, SAR-MAP can be immediately applied to identify structural variations driving natural product activity in cell lines and primary human cells.

Improving the quality of per-cell biological measurements has been a focus of cytometry over the last 20 years and has led to significant improvements in understanding cellular processes governing life, death, and specification of cell identity[1]. However, in preclinical drug discovery and chemical biology it is common to use target-based approaches, or alternatively, to deploy imaging or label free approaches that are neither single cell nor multiplexed, or lack in resolution or throughput. Though target-based drug discovery became a predominant approach after sequencing the human genome in 2001, only a 9.39% of FDA approved drugs have been discovered through target based approaches in contrast with the 77.9% discovered with phenotype-based approaches[2,3]. Phenotypic drug discovery is a target-agnostic approach centered on interrogating molecules in a biological system relevant to a disease of interest; this approach has been utilized to associate promising molecules with previously unknown molecular mechanisms, signaling pathways and targets[4]. Phospho-specific flow cytometry (phospho-flow) was developed to measure a range of cellular functions at the single cell level and has emerged as a high content, phenotypic drug discovery approach[5–8]. The technique involves quantifying one or more features of individual cells that have been stained with a panel of fluorescently tagged antibodies directed against specific epitopes, such as surface proteins, transcription factors, and phospho-proteins. Thus, phospho-flow can be leveraged to track signaling responses in clinically relevant cell subsets. The resulting multiparameter, single cell output provides many useful metrics to quantify compound effects compared to a single target or readout-based approach[2].

Throughput, consistency, and costs have been improved by a technique called fluorescent cell barcoding (FCB) that labels cells from a given well or sample with a unique signature of different levels of fluorescent dye - a barcode - so that they can be mixed, stained, and analyzed as a single sample[9] (Supplementary Fig. 1). Phospho-flow-based single cell profiling has been used to screen sets of small molecules and metabolite extracts and has identified pathway- and cell-selective inhibitors in cell lines and primary samples[6,10,11]. For instance, phospho-flow was used to evaluate the signaling profile for synthetic acylphloroglucinol scaffolds; here the platform was termed multiplexed activity profiling (MAP)[12]. These prior studies focused on sets of diverse molecules; thus, an underexplored opportunity is to apply multiplexed phospho-flow for connecting single cell signaling profiles to structural features as in structure-activity relationship (SAR) studies. Here, we introduce an experimental and computational platform for exploring structure-activity relationships using multiplexed activity profiling, which we will refer to as **SAR-MAP**.

While traditional chemical biology focuses on activity or selectivity against a single target, phospho-flow efficiently detects **bioactivity**, **selective activity**, and **signature profile** (see Supplementary Table 1 and below) in a single readout. Here, bioactivity refers to the ability of a molecule to elicit any biological response, as compared with a control. Notably, to detect

[1]Department of Cell and Developmental Biology, Vanderbilt University, Nashville, TN, USA. [2]Chemical & Physical Biology Program, Vanderbilt University, Nashville, TN, USA. [3]Department of Pediatrics-Neurology, University of Colorado Anschutz Medical Campus, Aurora, CO, USA. [4]Department of Chemistry and Center for Molecular Discovery (BU-CMD), Boston University, Boston, MA, USA. ✉e-mail: jonathan.irish@cuanschutz.edu

bioactivity, it is useful to measure a wide range of cell functions to capture diverse potential impacts on cell biology that a compound might have. In these studies, bioactivity was quantified using simultaneous per-cell measurement of hallmark cell functions, including apoptosis (c-CAS3[13]), DNA damage response (γH2AX[14]), membrane permeabilization (Ax700[7]), RTK signaling, mTOR pathway activity, and translation (p-STAT3[15], p-STAT5[16], p-ERK[17], p-AKT[18], p-S6[19,20], p-4EBP1[21]), T-cell receptor signaling (p-SFK[22]), proliferation and cell cycle (p-ERK[17], p-S6[19,20], p-HH3[23], Ki67[24]), cell size (forward light scatter), cell morphological complexity (side light scatter), and cell killing (cell count). For individual readouts, activity was further dissected with traditional cytometry approaches, such as log-like fold change in median signal intensity over vehicle in treated vs. control cells or differential percent of cells in a gate for treated vs. control cells (see *Methods*). These portions of the bioactivity profile are more like traditional measures of pharmacodynamic effect but still are likely measuring secondary effects rather than primary mechanisms after 16 hours of treatment. For a specific molecule, sub-class of molecules, or computational subgroup (cluster) of molecules with similar effects on cells, we can also refer to a signature profile of bioactivity when a specific combination of activities is repeatedly observed for a molecule or group. Finally, molecules might demonstrate cell-selective activity, which here means displaying activity in a cell type or cell subset. Pharmacodynamic effects such as potency in proximal target engagement (i.e., eIF4A1 clamping for rocaglates) were measured in prior studies and compared here to measured bioactivities and signature profiles.

We focused on quantifying bioactivity, selective activity, and signature profiles (example metrics for each in Supplementary Table 1). Notably, these bioactivity readouts are compatible with phenotypic screening approaches used in molecular discovery and untargeted approaches, since the goal is to broadly survey a wide range of biological space across diverse cell types rather than to assume activity specifically for impacting a particular molecule, pathway, or cell type. Here, we specifically used bioactivity as a starting point to identify a small molecule family for phospho-flow-based SAR analysis. This small molecule family, a natural product class called rocaglates (flavaglines), are highly potent inhibitors of cap-dependent translation, which are known to act by "clamping" DEAD box helicases such as eIF4A and DDX3 on polypurine-rich RNA sequences[25,26]. This mode-of-action has been shown to impede protein synthesis via multiple interconnected outcomes, including the prevention of ribosome recruitment, inhibition of 40S scanning, and depletion of eIF4A from the eIF4F complex[27]. Protein synthesis is an attractive cancer target as it is tightly regulated in normal cells but often highly dysregulated in cancer cells, leading to uncontrolled growth and survival[28]. As such, rocaglates have been cited for their ability to selectively induce cancer cell death while minimally disrupting healthy cells, specifically in leukemia[29,30]. What remains less understood, however, is how structural variations within the rocaglate chemotype impact the relative degree of inhibition of these known, as well as unknown, cellular processes in both cancer and healthy cells.

To probe this question, our approach draws on quantitative SAR approaches[31] combined with single cell bench techniques and modern data science tools that employ machine learning. Computational methods like t-SNE[32], UMAP[33], and PHATE[34] are used to analyze cytometry data[33] and are now growing in use in computational chemical biology[6,10,34]. Recently developed algorithms, such as Tracking Responders EXpanding (T-REX)[35], Marker Enrichment Modeling (MEM)[36], and Velociraptor[37] might be used to augment analysis and facilitate identification of rare cells and quantification of contextual protein enrichment. Tools like these that can resolve cellular heterogeneity and consider many readouts simultaneously may help to reveal connections between chemical structure and cellular bioactivity.

Overall, SAR-MAP reveals differences in bioactivity, signature profile, and cell type selective activity within rocaglate subclasses previously thought to be mechanistically homogenous. Notably, rocaglates with a pyrimidinone ring fusion (RPs) stand out as a subgroup that exclusively targeted leukemia cells and not healthy cells, with individual members displaying exceptional

ability to activate intrinsic DNA damage responses that lead to the selective destruction of malignant cells while retaining mechanistic target of rapamycin (mTOR) pathway activity. Further, three structural features were found to be correlated with this multidimensional signaling profile seen only in leukemia cells, with one confirmed in follow-up testing, suggesting that the unique structural features which separate the exceptional RPs from other members of this leukemia cell targeting subclass can be leveraged in the pre-clinical optimization and therapeutic translation of this chemotype. Overall, this work introducing the use of SAR-MAP helps show the value of single cell phospho-flow in characterizing large networks of fundamental cell signaling processes and simultaneously quantifying multiple chemical biology readouts.

## Results

### Rocaglates are highly bioactive in leukemia cells

A test of 600 diverse molecules from the Boston University Center for Molecular Discovery (BU-CMD) compound collection, with representation from 120 distinct chemotypes (**Diversity Set**, Fig. 1d), was conducted at 10 μM using phospho-flow in combination with FCB for multiplexing (Supplementary Fig. 1). The goal of this experiment was to identify a chemical family ('chemotype')[38] with exceptional leukemia cell bioactivity; therefore, MV411, a widely established model system for studying human acute myeloid leukemia (AML), was selected as the cell line for the Diversity Set screen.

The panel selected for this initial test included post-translational modifications of proteins representing four fundamental cell processes: c-CAS3, γH2AX, p-HH3, and p-S6 S235/S236 (Supplementary Table 2). The inverse hyperbolic sine (arcsinh) fold change in median fluorescence intensity (MFI) vs. vehicle was calculated for each compound across each of the four functional readouts (Fig. 1e)[10,11,18,35]. While few compounds decreased the fold change in MFI vs. vehicle for any readout, 65 molecules were found to increase p-S6, c-CAS3, and γH2AX significantly compared with vehicle (**Bioactive Set**, Fig. 1d). Of the 65 bioactive molecules selected, 13.8% (9/65) were rocaglates, a chemotype which comprised only 3.2% of the initial set of 600, and predominant in the bioactive set (Fig. 1f). The next most represented chemotype in the Bioactive Set, the guanidines, was around half as represented as the rocaglates (5 molecules or 7.69% of Bioactive Set) despite constituting a larger percentage of molecules in the original Diversity Set (23 molecules or 3.83% of Diversity Set). Thus, while other chemotypes displayed activity, the level of enrichment for bioactive rocaglates during this phase was exceptional across chosen members of all studied chemotypes.

Target- and diversity-oriented synthesis techniques have enabled the production of diverse rocaglate-inspired natural product analogues, including rocaglate subclasses both inspired by nature and discovered by serendipity. The BU-CMD houses a series of synthetic rocaglate structural variants[39] spanning three structural subclasses: 1) regular rocaglates (RR), exemplified by the founding natural product, rocaglamide A (RocA) (Supplementary Fig. 2a), silvestrol, and eliptifoline, containing a cyclopenta[*b*]-benzofuran scaffold and no other ring fusions, 2) rocaglate pyrimidinones (RPs), exemplified by natural products aglaroxin C (Supplementary Fig. 2c) and aglaiastatin, containing a mono- or bicyclic pyrimidinone ring system fused to the cyclopenta[*b*]-benzofuran core[40], and 3) amidino-rocaglates (ADRs), a rocaglate subclass not found in nature, which contain an amidine ring fusion to the cyclopenta[*b*]benzofuran core and have been shown to exhibit the most potent translation inhibition for any rocaglate to-date[27,41].

All 19 rocaglates in the Diversity Set were RRs. To investigate whether rocaglates from different structural subgroups were bioactive in MV411 leukemia cells and how this bioactivity extended to additional cell types, we selected a focused cohort (**Rocaglate Set**, Fig. 1d) of 37 rocaglates with representation from each of the three subclasses for testing using the SAR-MAP platform. SAR-MAP refers to testing **S**tructure **A**ctivity **R**elationships for a closely related class of molecules using the high dimensional, single cell, **M**ultiplexed **A**ctivity **P**rofiling assay and associated machine learning-based

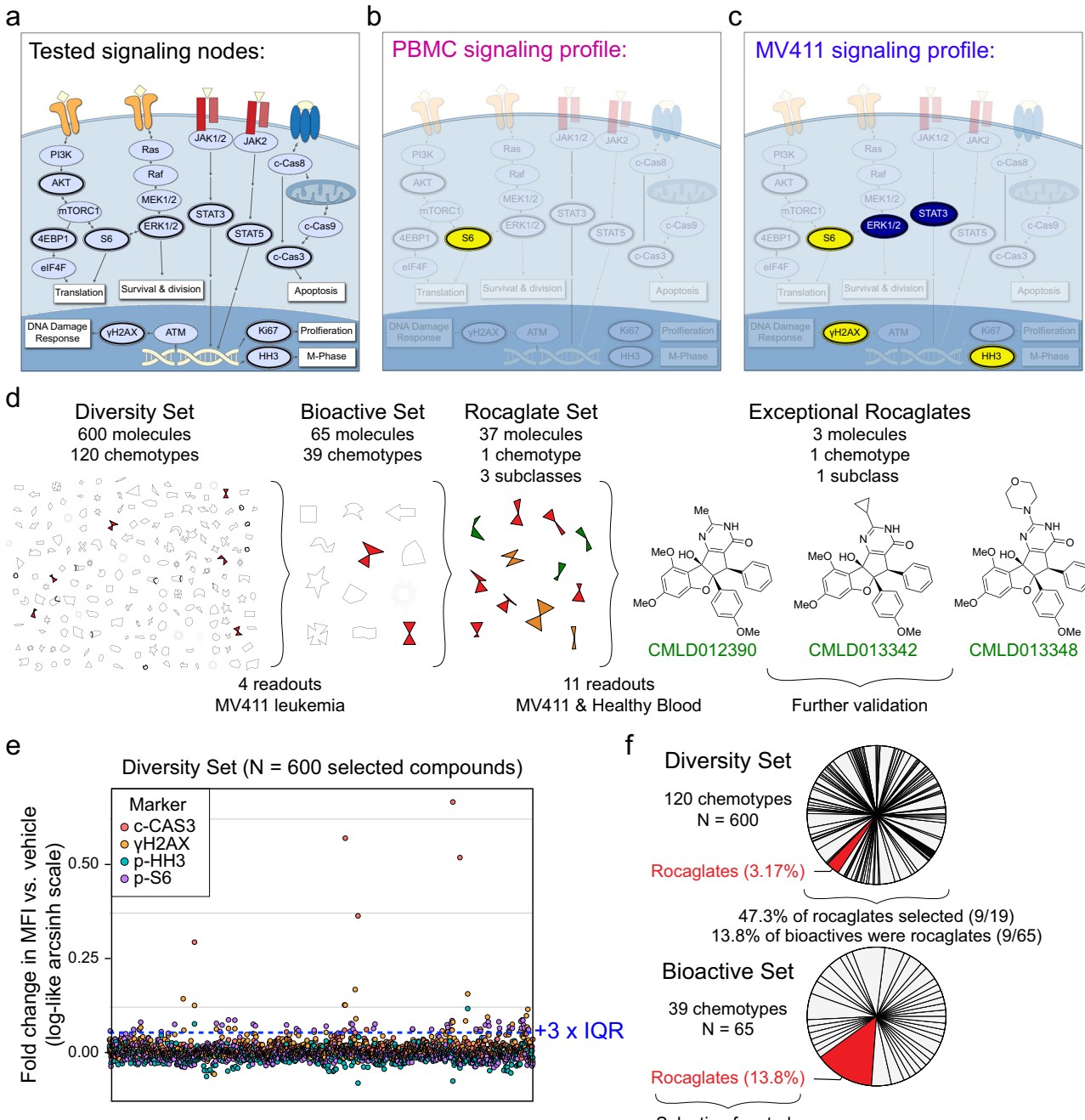

**Fig. 1 | Rocaglates displayed exceptional activity in a multiplexed single cell bioactivity assay. a** A model of a cell annotated with the key readouts measured (dark outlines). mTOR pathway activation is tested through p-AKT, p-ERK1/2, p-S6, and p-4EBP1, which regulates translation, among other cellular processes. Activation of the DNA damage response is explored through γH2AX. Ki67 and p-HH3 are indicative of proliferation and M-phase of the cell cycle, respectively. p-STAT3 and p-STAT5 are transcription factors that regulate the expression of cell cycle, survival, and pro-inflammatory genes. Activation of apoptosis is measured through the detection of c-CAS3. Key readouts activated (yellow) or inhibited (blue) by the exceptional three rocaglates (shown in **d**) are depicted for peripheral blood mononuclear cells (PBMC) in (**b**) and MV411 in (**c**). **d** A symbolic representation of the 600-compound Diversity Set comprised of 120 chemotypes, the 65 molecules identified as bioactive, and the

Rocaglate Set selected for structure activity relationships (SAR). The names and structures for the three exceptional rocaglates are shown. The number of molecules, chemotypes, and subclasses, where relevant, are depicted above each respective phase of testing. The number of readouts and cells used for testing are depicted below each phase. **e** The arcsinh fold change in median fluorescence intensity vs. vehicle for the 600-compound Diversity Set across the four readouts tested. Each readout is represented by a different colored circle. The bioactivity threshold was drawn at the vehicle median (0) + 3 × vehicle interquartile range (IQR) (0.0573). A total of 2,027,657 intact, single, live cells were collected in this experiment. **f** Pie charts depicting the proportion of each chemotype represented in the Diversity Set (top) and Bioactive Set (bottom). The pie slice corresponding to rocaglates is highlighted in red in each set. Here, N refers to the number of compounds.

analyses. The molecules, alongside vehicle in triplicate, were tested at 10 μM on MV411, H524 (human non-small cell lung cancer cells), mouse embryonic neural stem cells (eNSC), and healthy peripheral blood mononuclear cells (PBMC). The purpose of testing H524, eNSC, and PBMC in

addition to MV411 was to elucidate the cell type(s) most responsive to rocaglates and suitable for further analysis. The primary staining panel was selected to test a range of cell functions and has been validated in previous work: γH2AX, p-HH3, p-S6 S235/236, Ki67, p-S6 S240/244, p-STAT3, p-

STAT5, p-ERK1/2, p-4EBP1, p-SFK, and p-AKT (Supplementary Table 2)[42].

A heatmap of the log-like arcsinh ratio of the median phospho-protein signal in treated cells as compared to the signal in the vehicle (i.e., arcsinh [$MFI_{treated}$ / $MFI_{vehicle}$]) was performed for each marker and compound for each of the four cell types (Supplementary Fig. 3). Inspection of the heatmaps revealed that all four cell types were responsive to rocaglate stimulation, thus confirming rocaglates are a robustly bioactive chemotype across multiple model systems (Supplementary Fig. 3). Notably, while γH2AX activation was seen across most rocaglates, regardless of subclass in H524 and eNSC, γH2AX intensity appeared subclass dependent in MV411 and PBMC (Supplementary Fig. 3). The IQR of the intensity values for the three vehicle wells across all markers was computed and used as a metric of quality control alongside event counts for the four cell types. Given that MV411 and PBMC had both higher event counts and lower vehicle IQRs than H524 and eNSC, these cell types were selected for further analysis (Supplementary Fig. 3, Fig. 2a). Thus, further investigation of rocaglate bioactivity in MV411 was conducted to understand the diverse signaling patterns triggered across structurally different rocaglates. Then, rocaglates were characterized for their selective activity in targeting leukemia over healthy cells by contrasting bioactivity patterns in MV411 with PBMC.

First, the bioactivity across rocaglate structural subclasses was validated in MV411. A t-SNE plot was made using MV411 data from all 45 conditions tested (37 rocaglates, 3 vehicles, and 5 controls) and then subdivided based on the subclass of origin (and vehicle) for comparison of high-dimensional signaling profiles (Fig. 2b). T-REX was used to quantify differences in cell populations seen in the t-SNE plots for each of the three rocaglate subclasses. Specifically, the t-SNE from each of the three respective rocaglate subclasses was compared to the pooled set of the 3 vehicle wells (Fig. 2c). T-REX identified the cells that distinctly responded to vehicle-treatment (blue) and rocaglate treatment (red). For each subclass, T-REX degree of difference was computed to quantify how different the subclass-treated cells were from vehicle-treated cells, a bioactivity metric. All rocaglate subclass-treated cells had a T-REX degree of difference that was greater than 60%, indicating substantial change upon 16 hours of rocaglate treatment, relative to vehicle. In comparison, when T-REX was performed on vehicle 3 vs. the pooled set of vehicles 1 and 2, the T-REX degree of difference was <1%, indicating the vehicles were highly like each other. Therefore, the observation from the Diversity Set screen that the rocaglates displayed outstanding bioactivity against leukemia cells (Fig. 1e) was apparent in shifts in density on t-SNE (Fig. 2b) and in portions of the T-REX maps for each subclass displaying a difference of >95% from vehicle (Fig. 2c, dark red and dark blue). Rocaglates not only displayed consistently high bioactivity, but they also displayed a wealth of different signature profiles as evidenced by the contrasting t-SNEs that suggested that SAR analysis would be especially fruitful.

To quantify the differences in signature profiles for each member of a subclass and for each subclass as a whole, the total T-REX degree of difference was compared for each compound well (dot) and each family (box and whiskers) as compared to the values for the pooled set of the 3 vehicle wells (Fig. 2d). As a representative of this analysis, the T-REX plot for **CMLD013342** vs. vehicle is shown in Fig. 2d (remaining compounds are shown in Supplementary Fig. 4). Then, T-REX degree of difference was calculated for each of the 40 comparisons (37 rocaglates + 3 vehicles). All 37 rocaglates had a T-REX degree of difference above the designated bioactivity threshold. Taken together, these results indicated that all tested rocaglates were bioactive against MV411 leukemia cells.

### SAR-MAP reveals subclass-specific bioactivity patterns

To further explore signaling patterns within and across the rocaglate subclasses, the arcsinh [$MFI_{treated}$ / $MFI_{vehicle}$] was performed for each marker and compound in MV411. This resulted in values that range from -1.5 (blue) through 0 (no change, black) to +1 (yellow) on the arcsinh scale. Additionally, the fold change in cell count for treated wells as compared to

vehicle cell count was calculated (i.e., cell count$_{treated}$ / cell count$_{vehicle}$). Given that this dataset was gated for live, intact, single cells as part of pre-processing, cell count acts as a measure of basic cell killing. In Fig. 3a, b compounds were arranged according to rocaglate subclasses to allow comparison of marker intensities and cell counts within and between subclasses, respectively. In Fig. 3c, d marker intensities and cell counts were arranged by hierarchical clustering of the 11 marker intensity values (top of Fig. 3c). When organized by subclass, RPs displayed a more consistent signature than other subclasses (median interquartile range (IQR) across all markers for RP = 0.07, RR = 0.2, and ADR = 0.2), suggesting that the RPs are the most consistent subclass in terms of cellular mechanism (Fig. 3a). The RP signature profile, which was also observed in some RRs, included exceptionally strong γH2AX and inhibition of p-ERK, a signature indicating activation of DNA damage response and inhibition of MAPK proliferative signaling. In contrast to RPs, most ADRs inhibited γH2AX and p-4EBP1 and activated p-AKT and p-STAT5, a signature of cell survival signaling in leukemia cells that, while demonstrative of the ability of the approach to identify contrasting mechanisms, is not ideal for cancer research. RPs and ADRs had high respectively high and low cell counts relative to the vehicles. RRs were heterogeneous in both the pattern of phosphorylation seen following rocaglate treatment and cell counts; some RRs displayed patterns similar RPs and some to ADRs.

Grouping by unsupervised clustering of bioactivity led to the successful separation of RPs into clusters 2 and 3 and ADRs into clusters 4 and 5 (except for **CMLD012600**); RRs were interspersed throughout clusters 2–5 (Fig. 3c). Clustering was driven by differences in γH2AX; there was a greater than four-fold difference in γH2AX intensity between clusters 3 (mean log-like arcsinh transformed MFI: 0.47) and cluster 4 (mean log-like arcsinh transformed MFI: −0.83) (Fig. 3c). Structures for the 37 rocaglates grouped according to rocaglate subclass and hierarchical clustering can be found in Supplementary Figs. 5 and 6, respectively. Interestingly, cell count was positively correlated with γH2AX intensity; cluster 3 had both the highest γH2AX intensity and average cell count over vehicle (0.52) and cluster 4 had the lowest average cell count over vehicle (0.14) (Fig. 3d).

The next goal was to assess leukemia-cell selective activity using healthy human blood cells. The percentage of γH2AX positive (% γH2AX+) cells for each compound (and vehicle) was quantified for PBMC and MV411 (Fig. 4a). To distinguish compounds that triggered significant DNA damage in MV411 and PBMC, a threshold of vehicle median + 3 × vehicle IQR was first considered; this is the cutoff that was utilized in Figs. 1a and 2d. However, at this cutoff (7.0% γH2AX+), all except six rocaglates triggered significant DNA damage in MV411; this is another indication of assay quality and exceptional rocaglate bioactivity. To develop a more stringent threshold, a Wilcoxon rank sum test was conducted utilizing the % γH2AX+ values in MV411 and PBMC for the 37 rocaglates; an alternative hypothesis that the true median % γH2AX+ was greater than the median vehicle % γH2AX+ was utilized. The confidence interval generated by this hypothesis test, 13.3%, was used as the significance threshold. Only the vehicles (purple) and three RRs (red) did not trigger significant DNA damage in either cell type, further demonstrating the exceptional bioactivity of rocaglates (Fig. 4a). Nearly all ADRs (orange) and one RR (green) (**SDS-1-021**[43], in the set as both racemic (**CMLD011880**) and enantioenriched (**CMLD010508**) stocks) triggered significant DNA damage in PBMC alone; these rocaglates displayed selective activity for targeting healthy blood, again showcasing the power of the SAR-MAP platform to reveal diverse effects, including those desired and those not. All RPs (green) and some RRs (red) led to the desired leukemia-selective induction of γH2AX.

The log2 fold ratio of the percentage of γH2AX+ cells was next compared between MV411 and healthy blood (i.e., log2 [% γH2AX+$_{MV411}$/ γH2AX+$_{PBMC}$]) to quantify selective activation of the DNA damage response in leukemia cells (Fig. 4b). All RPs and the select RRs that demonstrated leukemia-specific induction of γH2AX in Fig. 4a were found to be in the top right quadrant, as they had maximal DNA damage induction in MV411 and a specificity of this induction to leukemia cells. Three RPs,

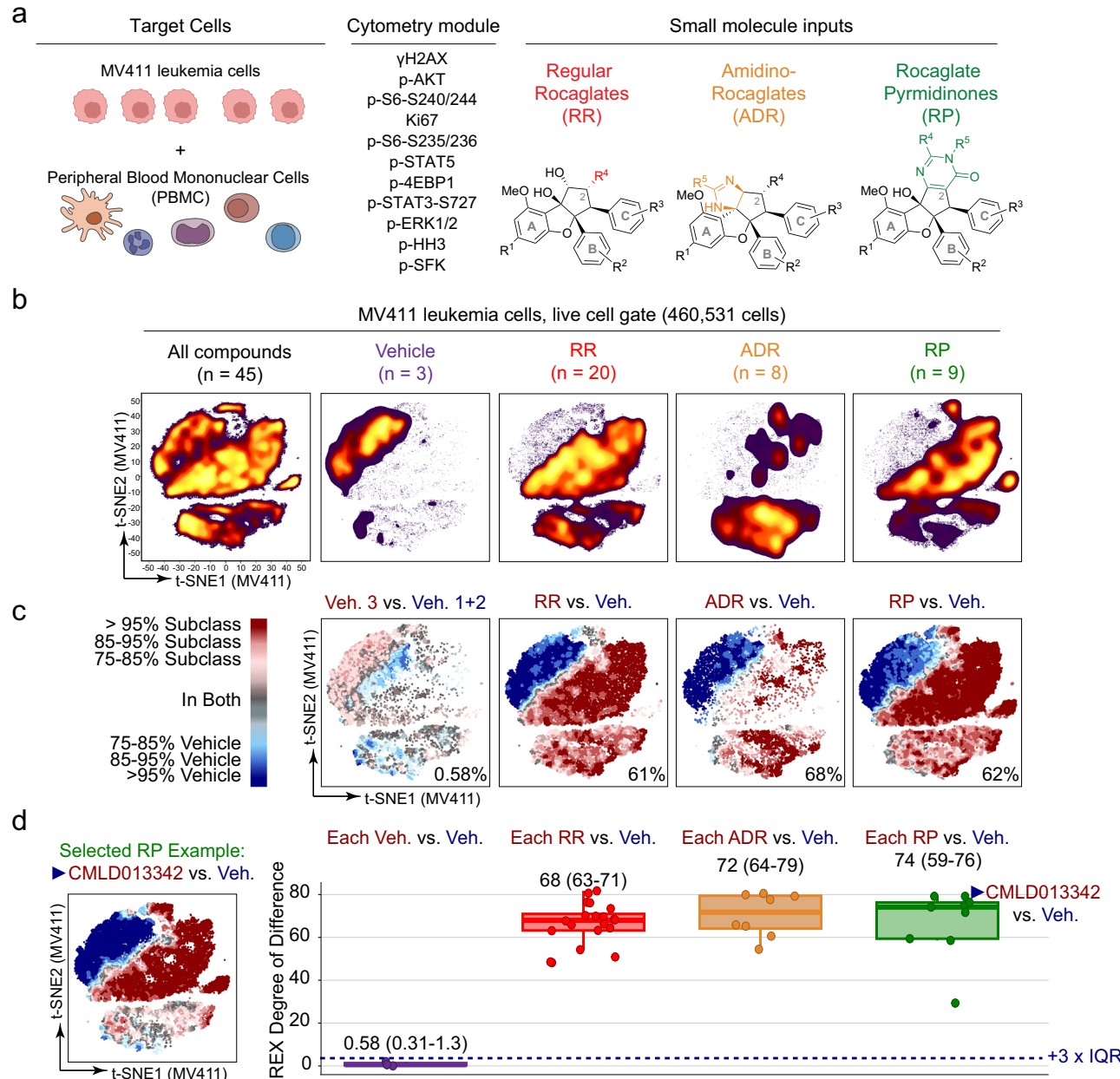

**Fig. 2 | Rocaglates are a chemical family with bioactivity against leukemia cells.**
**a** Shown, are three elements core to a single cell chemical biology experiment with flow cytometry – target cells, a module of cytometry readouts, and small molecule inputs - and the inputs selected for these studies. MV411 and PBMCs were selected as the set of target cells. Eleven readouts of core cell functions were selected for the cytometry module. Lastly, rocaglates from the three depicted structural subclasses were chosen as the small molecule inputs. **b** Plot depicting the result of performing a t-SNE analysis on the entire pre-processed MV411 dataset (All compounds) and dividing based on the category of input (Vehicle, RR, ADR, and RP). Dots are colored based on cell density. **c** Tracking Responders EXpanding (T-REX) plots depict regions of significant difference between the t-SNE of one rocaglate subclass vs. vehicle-treated cells in MV411. As a representative of the variation between the three tested vehicle wells, a T-REX plot is shown for vehicle 3 vs. pooled vehicles 1 and 2 (leftmost plot). The T-REX degree of difference [(# red cells + # blue cells) / total #

cells] for each analysis is reported in the lower right corner of each plot. Axes scales are as in (**b**). **d** Box and whisker plot of T-REX degree of difference for each compound vs. vehicle-treated cells in MV411 grouped and colored according to structural subclass. Box-plot elements: center line = median; box limits = upper and lower quartiles; lower whisker = smallest observation greater than or equal to lower quartile - 1.5 × interquartile range (IQR); upper whisker = largest observation less than or equal to upper quartile + 1.5 × IQR; all data points are shown. The n or number of independent compounds utilized for each box is listed in (**b**) above and in the color corresponding to the box. Median, quartile 1, and quartile 3 T-REX degree of difference values for each subclass are provided above. A dashed navy line is shown at 3.53% to indicate a significance threshold [vehicle median (0.583%) + 3 × vehicle IQR (2.94%)]. Representative T-REX plot for CMLD013342 vs. vehicle is shown to the left of the box and whisker plot for reference. T-REX plots for the remaining compounds are depicted in Supplementary Fig. 4b.

**CMLD012390, CMLD013342**, and **CMLD013348** had the highest log2-fold ratio of % γH2AX+ in MV411 to % γH2AX+ PBMC. Again, RPs were the exception, this time in selective activity, as they were the only subclass with all members demonstrating a deadly leukemia-specific DNA damage response.

To gain further insight into the cell subpopulations with high γH2AX intensity, biaxial plots of side scatter vs. γH2AX were generated for representative rocaglates from each quadrant in Fig. 4b (Fig. 4c). Side scatter (SSC) is a measure of orthogonal laser light scatter that typically increases with cell granularity or membrane complexity[44]. PBMCs include

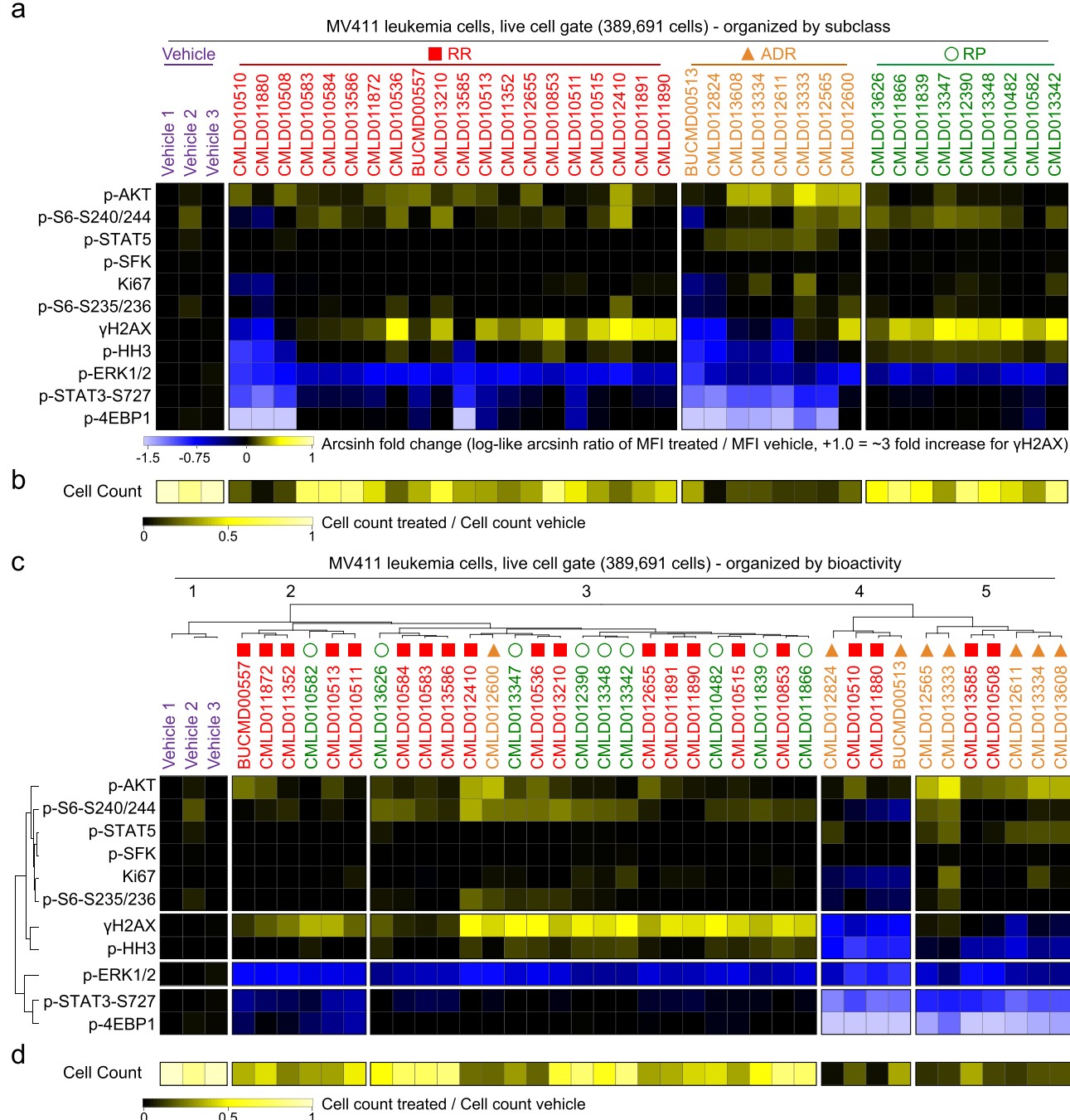

**Fig. 3 | Rocaglate subclasses had distinct patterns of bioactivity. a** Heatmap depicting the arcsinh ratio of the median fluorescence intensity for each compound (listed on top of heatmap) and readout (listed left of heatmap) by the median fluorescence intensity of Vehicle 1. Cells on the heatmap range from light blue for the lowest values to bright yellow for the highest values. Compounds are grouped and colored according to the rocaglate subclass listed on the top of the plot. **b** Heatmap of the cell count for each compound divided by the cell count for Vehicle 1. Cells on this heatmap range in color from black, for molecules with low cell count relative to vehicle, to yellow for molecules with a similar cell count to vehicle. Heatmap is clustered according to rocaglate structural subclass as displayed in (**a**). **c** Heatmap as in (**a**) clustered according to a dendrogram of the transformed median fluorescence intensity for each compound and readout. **d** Heatmap as in (**b**) clustered according to a dendrogram displayed in (**c**).

lymphocytes such as T cells, B cells, and NK cells and myeloid origin cells such as monocytes. In healthy blood, >95% of lymphocytes have low SSC due to being small, non-complex, and in G0 phase (i.e., not replicating). Myeloid origin PBMCs have high side scatter due to complex membrane and organelle structures[45,46]. Thus, the cells responsible for γH2AX activation in PBMC – those in the bottom two quadrants – were SSC low and identified as healthy lymphocytes (Fig. 4c). Because MV411 have a clonal origin, differences in SSC likely arise from variation in morphological responses to compound stimulation and differences in epigenetic states and not due to differences in cell type or genetics. For instance, cells in the early stages of apoptosis have high SSC[47]. For **CMLD12600**, the γH2AX high cells had mid to high side scatter. For **CMLD013342**, there were γH2AX high cells with mid to high side scatter in addition to a population of cells with exceptional γH2AX, that generally had higher side scatter.

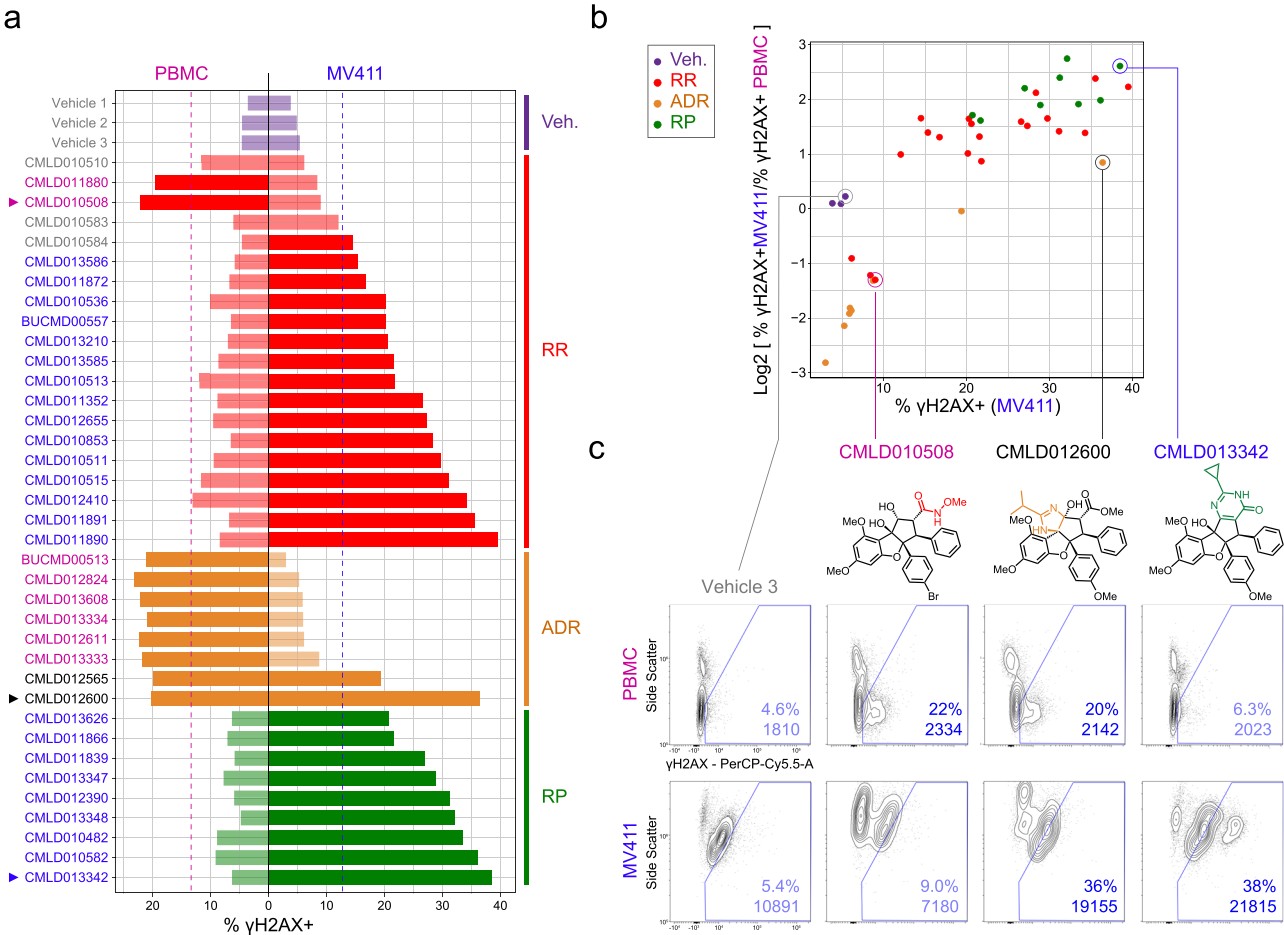

**Fig. 4 | MV411 specific activity was observed in the rocaglate pyrimidinone subfamily and some regular rocaglates. a** Bar plot of percentage of γH2AX positive (% γH2AX+) cells for each compound grouped and colored according to rocaglate subclass listed on the right side of the plot. Bars on the left side of the vertical black line correspond to the compound response in peripheral blood mononuclear cells (PBMC) and the right side corresponds to the compound response in MV411. The dotted vertical pink and blue lines correspond to a significance threshold of 13.3% γH2AX+ cells in PBMC and MV411, respectively. Compounds that cross the threshold have darkened colored bars. The compound name (or vehicle) is listed on the far left side of plot colored according to the following system: blue = increases % γH2AX+ cells past threshold in MV411 and not in PBMC, pink = increases % γH2AX+ cells past threshold in PBMC and not in MV411, light gray = does not increase % γH2AX+ cells past threshold in either cell type, dark gray = increases % γH2AX+ cells past threshold in both cell types. Arrows are displayed to the left of compounds that are shown in (**c**). **b** Scatter plot of % γH2AX+ cells in MV411 on the x-axis and the log2 fold ratio of % γH2AX+ cells in MV411 to % γH2AX+ cells in PBMC on the y-axis. Each dot corresponds to a compound colored according to rocaglate structural subclass. Compounds that will be shown in (**c**) are circled and labeled. **c** Contour plots of Side Scatter vs. γH2AX for Vehicle 3, **CMLD010508, CMLD012600,** and **CMLD013342**, respectively from left to right in PBMC (top) and MV411 (bottom). The contour plot was drawn with 10% of cells per contour and outliers starting at 10%. The blue line indicates the expert drawn γH2AX+ gate. The % γH2AX+ cells within the gate and the median fluorescence intensity are written in blue in the lower right corner. Corresponding structures for each compound are depicted below the compound name.

## The RP subclass has a leukemia-selective signature profile

An advantage of SAR-MAP is the ability to characterize cell subpopulations. One prominent population of interest was apparent after treatment with RPs. To characterize the signature of these cells, the population was first identified using T-REX. T-REX analysis compared cells treated with compounds from each of the three respective rocaglate subclasses to a pooled set of cells treated by members of the other rocaglate structural subclasses (including vehicle) (Fig. 5a). T-REX identified a subpopulation of cells that was only induced in response to RP treatment (RP island, Fig. 5a). Coloring the grouped t-SNE for all 45 compounds based on measurements for each functional readout showed that the signature profile of the RP island was high p-4EBP1 and γH2AX (Supplementary Fig. 7). Thus, the signature profile included preservation of **m**TOR **a**ctivity during a **D**NA **D**amage response and was named the **MADD signaling profile**. This profile was distinguished from common DNA damage responses

because γH2AX was activated in cells lacking phosphorylation of 4EBP1 and other phospho-proteins (i.e., the mTOR pathway was still active).

To test whether this signature profile was a feature of RPs, the percentage of cells with the MADD signaling profile was calculated for each rocaglate (Fig. 5b, Supplementary Fig. 8). A Wilcoxon Rank Sum test was calculated for RPs vs. the remaining subclasses and indicated that the MADD signaling profile was statistically significantly enriched for RPs ($p < 0.02$). A Marker Enrichment Modeling (MEM)[36] analysis was performed on cells with the MADD signaling profile to characterize contextual protein enrichment (absolute MEM labels range from 0 = no enrichment to +10 = highest enrichment) (Fig. 5c). The MEM protein enrichment for γH2AX and p-4EBP1 confirmed the MADD signaling profile of the RP island.

To assess whether RPs also activate the MADD signaling profile in healthy cells, MV411 and PBMC cells were combined into one

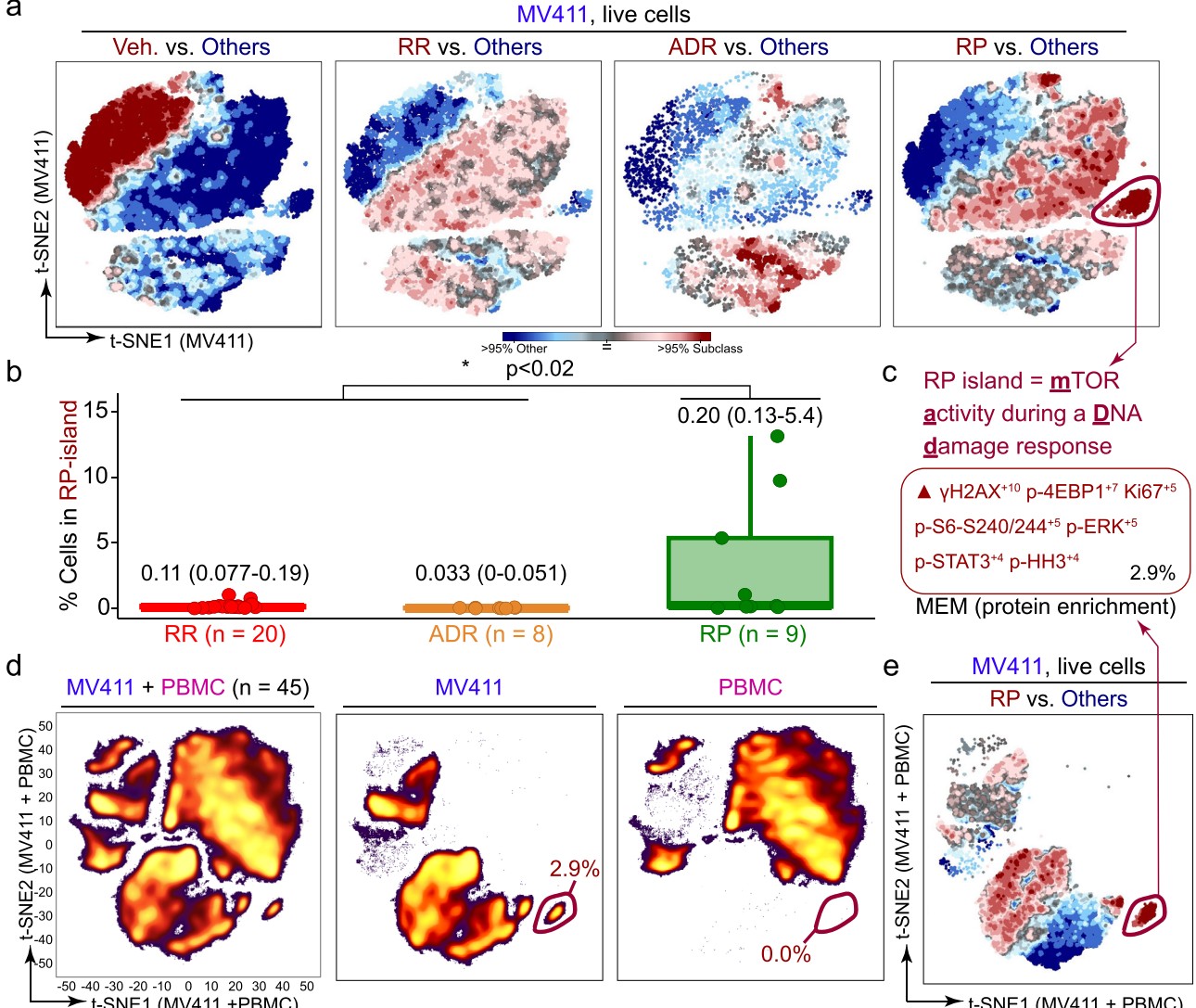

**Fig. 5 | The rocaglate pyrimidinone subfamily had a distinct, leukemia-specific population of cells with high DNA damage response and metabolic activity.**
**a** Tracking Responders EXpanding (T-REX) plots depict regions of significant difference between the t-SNE of one rocaglate subclass vs. the pooled set of all three remaining subclasses (including vehicle) for MV411 data. The rocaglate pyrimidinone (RP) island is circled in red in the "RP vs. Others" plot. Axes scales are as in Fig. 2b. **b** Boxplot depicting the percentage of cells in the RP island for MV411 for each rocaglate grouped by rocaglate subclass. Box-plot elements are as in Fig. 2d. The n or number of independent compounds utilized for each box is listed directly below the box. Median, quartile 1, and quartile 3 T-REX degree of difference values for each subclass are provided above. A Wilcoxon rank sum test of RP vs. remaining subclass % in gate was conducted and generated a p-value of 0.015 and a location shift of -0.14. Wilcoxon Rank Sum was selected to test whether two distributions have the same median. Flow cytometry measurements are not normally distributed, and the

field uses non-parametric tests; only one hypothesis was tested. Asterisk indicates that generated p-value was below the significance threshold of 0.05. **c** MEM protein enrichment label generated for the RP island of cells. The percentage of the total population of cells for the RP island is depicted in the lower right corner. **d** t-SNE plot of the MV411 and peripheral blood mononuclear cells (PBMC) data on a new embedding with corresponding x and y axes (left). The t-SNE plot for both cell types was divided into separate plots for MV411 (middle) and PBMC (right). The percentage of cells in an MV411-specific island is circled in red and depicted in the lower right of the MV411 and PBMC plots. A total of 1,108,887 intact, single live cells were considered in this analysis. **e** T-REX plot of RPs vs. the pooled set of all three remaining subclasses (including vehicle) for the MV411 data using the t-SNE performed on the MV411 and PBMC data together shown in (**d**). The MV411-specific population circled in red was identified to have the MEM signature depicted in (**c**). Axes scales are as in (**d**).

analysis, which included generating a new t-SNE embedding with corresponding x and y axis scales (Fig. 5d). After dimensionality reduction with t-SNE, cells were separated based on origin (MV411 or PBMC) (Fig. 5d). The MV411 t-SNE analysis revealed a distinct population of cells from the remaining cell density on the lower right side of the plot that was >99% specific to MV411. A MEM label analysis on this leukemia-specific population confirmed that the protein enrichment was identical to the label displayed in Fig. 5c (Fig. 5e). Thus, the RPs were the only subclass that had any cell subset (>1%) that was specific to that class in the MV411 cell line.

**Structural commonalities are seen for RPs with MADD profile**
To identify structural determinants of the MADD signaling profile, the t-SNE map of pooled RPs was subdivided based on compound of origin, and the percentage of total cells with the MADD signaling profile was quantified for the three vehicles, 2 controls (etoposide and nocodazole), a representative ADR (**CMLD013608**) (Fig. 6a), and all 9 RPs (Fig. 6b). The three RPs identified in Fig. 4b, **CMLD012390, CMLD013342**, and **CMLD013348**, and etoposide were the only compound treatments leading to greater than 5% of cells with the MADD signaling profile. Distinctly, **CMLD012390, CMLD013342**, and **CMLD013348**, commonly possess

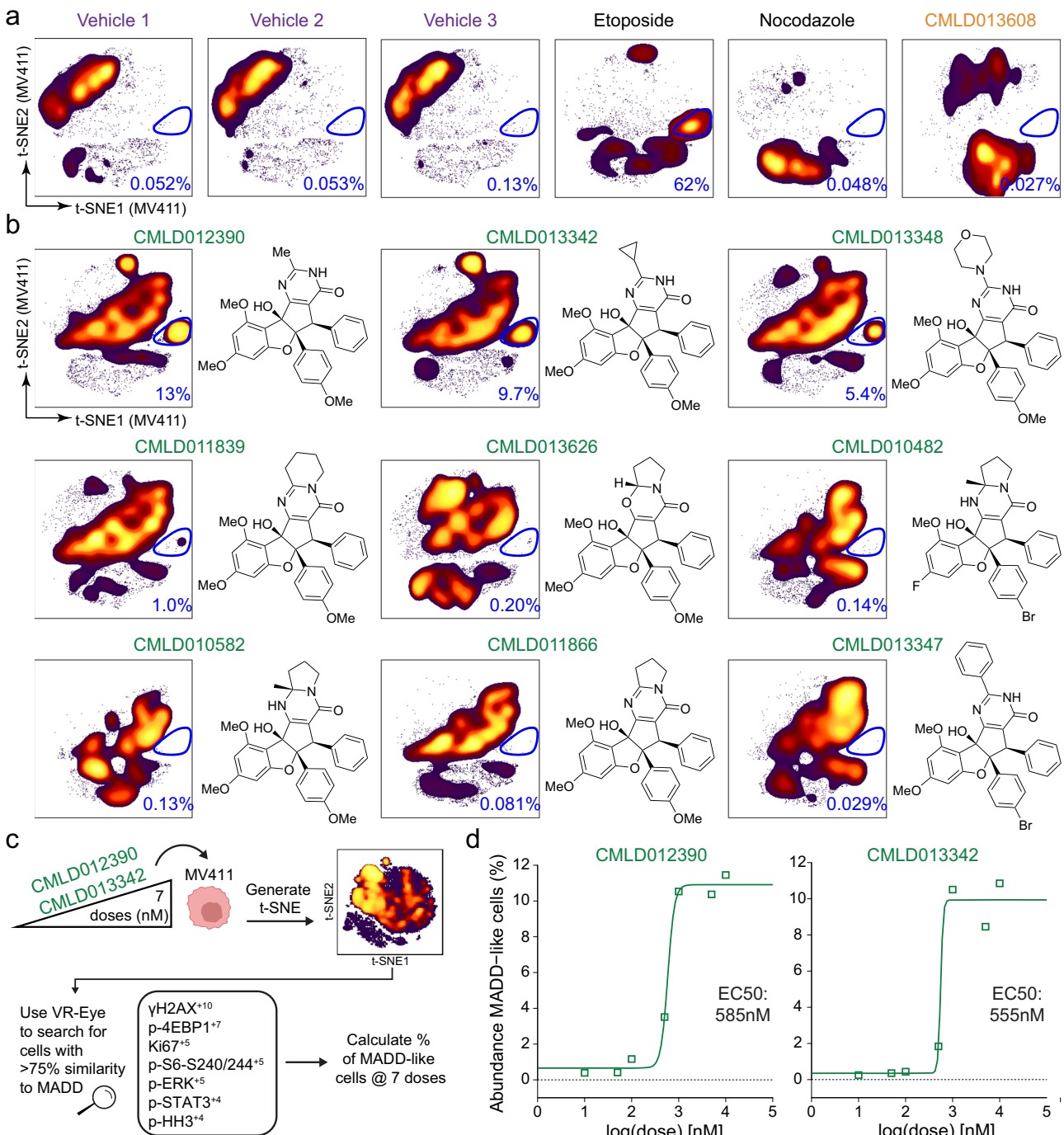

**Fig. 6 | Rocaglate pyrimidinones with mTOR activity during a DNA damage response possessed structural commonalities. a** t-SNE from Fig. 2b of all compounds in MV411 divided based on rocaglate well of origin is shown for the three vehicle wells, etoposide, nocodazole, and a representative amidino rocaglate (ADR) (**CMLD013608**). The percentage of cells in the mTOR activity during a DNA damage response (MADD) signaling profile gate circled in dark blue is included at the bottom right of each plot. Axes scales are as in Fig. 2b. **b** t-SNE from Fig. 2b of all compounds in MV411 divided based on rocaglate well of origin is shown for the set of 9 RPs in order of decreasing percentage of cells with the MADD signaling profile (t-SNEs for all individual rocaglates are shown in Supplementary Fig. 4a). The percentage of cells

with the MADD signaling profile is included at the bottom right of each plot. The chemical structure corresponding to each rocaglate is depicted on the right side of each t-SNE plot. Axes scales are as in Fig. 2b. **c** Schematic depicting dose response experimental workflow and analysis. A total of 249,258 and 257,576 intact, single, live cells for analysis for **CMLD012390** and **CMLD013342**, respectively, were utilized to generate the t-SNE in this workflow. **d** Dose-response titration curves depicting the abundance of MADD-like cells vs. log[dose(nM)]. A half-maximal activating concentration (EC50) curve as fitted to the data and EC50 values are shown in the middle of each plot. An EC50 of 585 nM ($p < 0.001$) was generated for **CMLD012390** and EC50 of 555 nM (p = n.s., $\alpha = 0.05$) was generated for **CMLD013342**.

three unique structural features that are found in isolation in the other six RPs, specifically a non-aryl pyrimidinone ring substituent (termed the RP "R-group"), a 4′-methoxy substituent on the rocaglate "B"-ring, and a monocyclic, N-unsubstituted pyrimidinone ring. Thus, three structural

features were identified as connected with the MADD signaling profile after investigation of the structures and signature profile at the compound level.

A dose-response experiment was performed in MV411 as an assessment of the reproducibility of MADD signaling profile activation by the

**Fig. 7 | Shift from 4' methoxy to 4' bromine dramatically shifted signature profile of exceptional rocaglate pyrimidinones.** A histogram overlay compares the distribution of marker intensity for four rocaglate pyrimidines tested on MV411 leukemia cells at 10 μM for 16 hours. Marker intensities for p-4EBP1, p-S6, and c-CAS3 are scaled by the log-like arcsinh ratio of the median fluorescence intensity (MFI) of the treated well by the minimum MFI for each marker and are colored according to the legend below each respective marker. The top set of compounds (**CMLD012390** and **BUCMD00617**) both have a methyl R-group and the bottom set of compounds (**CMLD013342** and **BUCMD00616**) both have a cyclopropyl R-group. Compounds within each set are identical other than the 4' substituent on the rocaglate B ring (methoxy vs. bromine). A total of 44,097 intact, single, live cells were collected in this experiment; 25,810 cells of these cells displayed as part of this figure.

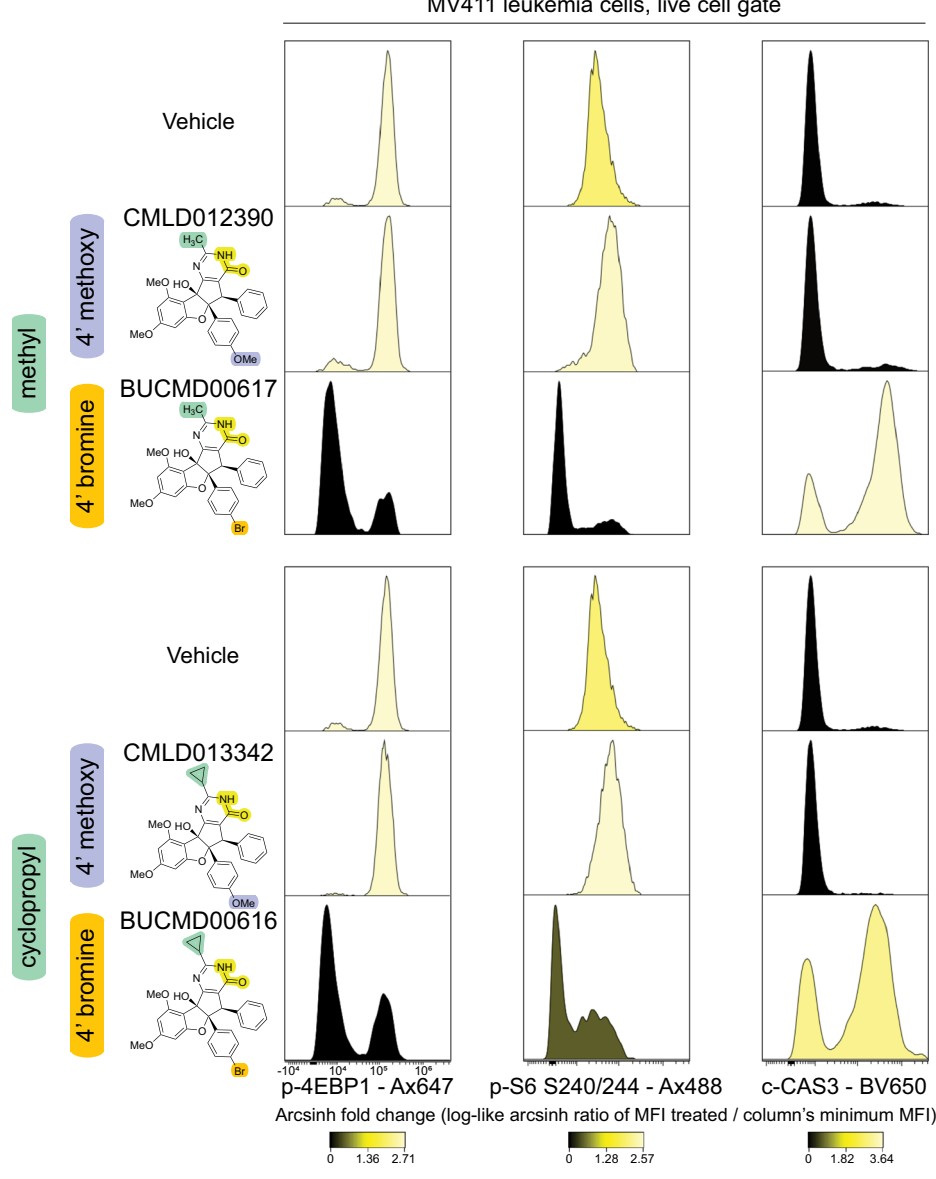

exceptional RPs. The top two exceptional RPs, **CMLD012390** and **CMLD013342** were chosen for testing alongside **CMLD013608**, a representative ADR, as a control (Fig. 6c, Supplementary Fig. 9a). Velociraptor-Eye (VR-Eye), a recently developed cell identification algorithm, was then used to identify the cells with the MADD signaling profile within the dose response dataset (Fig. 6c, Supplementary Fig. 9b)[37]. The resulting cells with greater than 75% similarity to the MADD signaling profile MEM label were used to calculate the half-maximal activating concentration (EC$_{50}$) for each of the two RPs (Fig. 6c). **CMLD012390** and **CMLD013342** had similar EC$_{50}$s of 585 nM and 555 nM, respectively (Fig. 6d). Thus, these two RPs demonstrated a reproducible, dose-dependent induction of this signature profile.

To assess the importance of the 4'-methoxy substituent on the rocaglate "B"-ring, phospho-flow was performed on the top two exceptional RPs, **CMLD012390** and **CMLD013342**, alongside two identical compounds with 4'-bromine substituents in place of 4'-methoxy substituents. While **CMLD012390** and **CMLD013342** demonstrated comparable p-4EBP1 and p-S6 240/244 intensities to vehicle and to one another, their respective brominated counterparts had on average 16-fold and 12-fold lower intensities for each marker, respectively (Fig. 7). Given that p-4EBP1 and p-S6 240/244 are downstream effectors of mTORC1, the 4'-methoxy substituent was identified as mechanistically significant for the MADD

signaling profile. Thus, SAR-MAP can be used to characterize differences in cell signaling patterns across a set of related molecules and determine the structural features that are responsible for dictating these contrasting signature profiles.

## Discussion

An advantage of single cell chemical biology approaches is the potential to simultaneously assess multiple readouts and functions, including bioactivity, signature profile, and selective activity. Thus, phospho-flow based multiplexed activity profiling can be used to include a range of cells that will be present in the human tissues in which compounds will need to function in vivo. Here we report a test of diverse compounds facilitating SAR based on multiplexed activity profiling, with the latter portion of this pipeline termed SAR-MAP. This revealed rocaglates, a rich class of polycyclic natural products derived from trees of the genus *Aglaia*, as a highly bioactive chemotype. Subsequently, all rocaglate pyrimidinones and most amidino-rocaglates tested were identified as structural subgroups that respectively selected for leukemia or healthy blood cells.

Since phospho-flow was first combined with FCB for chemical biology screening[6], this approach has been used for screening and characterization of promising compounds based on cell type- and pathway-specific effects. The screen of small molecules described here was similarly performed in

combination with FCB for multiplexing and used a focused panel of four functional states of proteins to efficiently profile the modulation of relevant pathways involved in leukemia cell translation and death. The screening set of 600 structurally diverse compounds is a relatively small library compared to what might be possible if this approach were scaled; part of the success of the study is likely due to the curated library and selection of key functional signaling events to measure as bioactivity readouts.

While previous studies have focused on selection of individual hits for follow-up testing, this approach was focused instead on identifying promising chemotypes. Of the 65-compound bioactive set, nine of the compounds were rocaglates, a natural product family previously thought to be well characterized for promising anti-cancer and anti-infective activities. Powered by AI and combinatorial chemistry, early drug discovery has seen a shift toward seeking out ways of expanding the chemical readout space explored in high throughput screening[48]. Rather than increasing chemical readout space, the screen presented here demonstrates how larger biological readout space can facilitate informative screens with focused sets of molecules. A challenge in virtual high throughput screens is obtaining a sufficiently large, high-quality test set to train the machine learning-based models; multi-dimensional, single cell approaches might provide higher-quality predictions with fewer molecules.

This study was the first phospho-flow-based test of rocaglates, to the best of our knowledge, and one of only a few that have characterized the bioactivities of structural subclasses within the chemotype[49]. Rocaglates have primarily been shown to have two key classes of DEAD-box RNA helicase targets: the three eIF4A homologs, and more recently DDX3[25,26]. While eIF4A is implicated in translation initiation, DDX3 is involved in a range of functions related to RNA metabolism[50]. Larger screens of >200 structurally diverse rocaglates have mainly monitored clamping of the three eIF4A homologs (eIF4A1-3) to RNA (i.e., proximal target engagement in Supplementary Table 1), in vitro translation, and cytotoxicity[27,41,51]. This test set of 37 rocaglates from three structural subclasses profiled multiple metrics of bioactivity, selective activity, and signature profile using 11 functional states of proteins across MV411 and PBMC cells.

Cytometric profiling of the 37 rocaglates in combination with t-SNE and T-REX enabled us to establish that these molecules were all bioactive against leukemia cells. Additionally, investigation of fluorescence intensity for each molecule and readout alongside cell counts enabled rocaglate bioactivity within and across subclass to be compared. An ideal subclassification scheme aims to identify the specific set of structural characteristics that best capture contrasting bioactivities. Current rocaglate structural subclasses captured many similarities in pathway readouts within each subclass and differences across subclasses. However, organizing the rocaglates by bioactivity using hierarchical clustering led to different subgroups of molecules. ADRs were separated from RPs, with RRs interspersed throughout the ADR and RP clusters. Taken together, this suggested that the set of structural characteristics that previously defined the rocaglate structural subclasses did not best capture differences in biological activity.

Though analysis of γH2AX via microscopy-based approaches is considered to be the most sensitive method of detection, it is time consuming and operator-dependent[52]. Flow cytometry has been validated as a reliable and sensitive measure of γH2AX and is more efficient, automated, and can be detected alongside multiple additional readouts[53]. Indeed, γH2AX was a readout that strongly contrasted across rocaglates in both PBMC and MV411 and provided further insight into differences in bioactivity and selective activity between structural subclasses. While some RRs and all RPs activated γH2AX in MV411 and not PBMC, ADRs only activated γH2AX in PBMCs. For RRs these findings are not surprising; rocaglamide A, an RR, has demonstrated leukemia-specific bioactivity in previous studies[54]. However, ADRs have been found to be exceptionally potent in vitro RNA clampers of eIF4A1 and eIF4A2[41]. RPs, the subfamily demonstrating uniform leukemia-specific induction of γH2AX, have primarily been reported as agents for hepatitis C virus and to show a possible bias for the inhibition of viral entry over translation inhibition[40]. The natural product aglaroxin C

(**CMLD011866**), an RP, has been cited as a potent cytotoxic agent against multiple cancer cell lines, though not yet as selective for cancer cells as opposed to healthy cells[55,56]. These findings were the first to our knowledge to identify leukemia-specific bioactivity, or a lack thereof, as a feature of RPs and ADRs, respectively.

By leveraging t-SNE, T-REX, and MEM, we were able to pinpoint and characterize a subset of cells with a signature profile including an activated DNA damage response and preserved metabolic activity. This MADD signaling profile was only present in RP and etoposide stimulated MV411 cells. Given that this population was <3% of total MV411 cells, it is likely that this would have been overlooked by approaches only accounting for bulk cell responses. Notably, MV411 is a clonal cell line isolated from the blasts of a 10-year male with biphenotypic B-myelomonocytic leukemia[57]. Given the assumption that all cells within MV411 are genetically identical and should thus respond homogenously to a given stimulus, it is peculiar for only a subset of cells to have the distinct MADD signaling profile. Though cell lines are core tools in cancer research, the assumption of clonality and functional homogeneity has been challenged in recent years. Genetic heterogeneity has been proven in hundreds of established cancer cell lines due to a variety of factors including evolving abundance of subclones, emergence of new genetic variants, epigenetic variation, and environmental stress; this heterogeneity can lead to differential sensitivity to molecules due to underlying diversity in gene expression patterns[58–60]. Therefore, identifying that only a subset of MV411s demonstrate the MADD signaling profile is not surprising, and further underscores the importance of approaches that can detect functional heterogeneity within cell lines. Further multiplexed flow cytometry studies that include cell surface markers, single cell RNA-sequencing, and validation on patient-derived xenografts might be performed to elucidate drivers of this variation in drug response.

Further investigation allowed us to uncover three exceptional RPs, **CMLD012390, CMLD013342**, and **CMLD013348**, with the MADD signaling profile. This response was not seen by aglaroxin C (**CMLD011866**) though its cytotoxic activity is warranted based on prior literature[55,56]. γH2AX acts as a sensitive marker for DNA double-stranded breaks (DSBs). Given that γH2AX activation is an early sign of the DNA damage response, the positive correlation with cell count seen at 16 hours is not surprising. Many existing cancer therapies such as etoposide and mitoxantrone act by introducing sufficient DSBs to activate cell death pathways[61]. 4EBP1 is a translational repressor protein; its phosphorylation by mTOR enables cap-dependent translation to proceed[62]. Therefore, 4EBP1 phosphorylation has primarily been associated with poor prognosis in AML[63]. However, recent publications have found that pharmacological suppression of mTOR activity in cancer cells can cause chemoresistant cell populations to persist through regulation of autophagy and G2/M cell cycle arrest; mTOR activation can increase chemosensitivity and predict better survival[64,65]. Moreover, etoposide had the greatest percentage of cells with the MADD signaling profile of all molecules tested; however, there were apparent differences in the t-SNE maps of etoposide in comparison with **CMLD012390, CMLD013342**, and **CMLD013348**. Therefore, the ability of the exceptional RPs to activate the MADD signaling profile should be explored further as a mechanism of inducing sustained cancer death alone or in combination with other therapies.

The compounds **CMLD012390, CMLD013342**, and **CMLD013348** have not been previously reported as distinct members of the RP subfamily. Notably, these compounds possess some structural similarities; they commonly have a B-ring methoxy group, a non-aryl R-group, and a monocyclic, N-unsubstituted pyrimidinone with unique steric properties that, in contrast to *N*-substituted RPs such as **CMLD011866**, is capable of equilibrating to an aromatic hydroxypyrimidine tautomeric form. Follow-up SAR-MAP confirmed that a shift from a 4′-methoxy substituent to a 4′-bromine led to decreased p-4EBP1 and p-S6 S240/244 activity and c-CAS3 activation. This marked shift in signaling profile detected via phospho-flow highlights an

advantage of multiparameter assays versus performing multiple sequential single readout assays for accelerating medicinal chemistry efforts. Having a larger biological resolution not only enables the impact of small chemical changes to be more efficiently discerned but reveals mechanistic information about the signal nodes simultaneously perturbed. Here, this includes the added knowledge that compounds with a 4′-methoxy substituent activate a DNA damage response without initiating apoptosis or suppressing mTOR pathway activity.

While the Diversity Set and Rocaglate Set screens were only performed once in each cell line, the single cell resolution of phospho-flow data offers some built-in replication in comparison with a platform solely detecting bulk cellular responses. Due to the inherent variability of cells, even within a clonal population, future studies should include replication of the Rocaglate Set screen on MV411, other leukemia cell lines, and primary leukemia samples to validate the contrasting bioactivity and signature profiles across structural subclass.

The prominent result of cooccurring γH2AX, p-4EBP1, and p-S6 S240/244 activity from the top two molecules identified - **CMLD012390** and **CMLD013342** - replicated in the two additional studies presented, thus providing encouraging evidence that the SAR-MAP platform is reliable and reproducible. This signature profile for **CMLD012390, CMLD013342**, and **CMLD013348** may imply an as-yet unknown target; alternatively, it is also possible that differences in potency or timing of engagement with eIF4A1 and/or DDX3 drive this unique profile. Thus, future experiments should include a time course and testing on primary leukemia with cell surface markers to resolve the cellular mechanisms and subpopulations driving the MADD signaling profile.

Critically, an expanded biological readout space enabled the result of small chemical changes to be efficiently resolved within curated molecular sets chosen to address SAR. SAR-MAP revealed rocaglate pyrimidinones (RPs) as a class with exceptional, leukemia-specific bioactivity. The 2-phase approach included testing 600 compounds representing diverse chemotypes followed by 37 family members that allowed within-chemotype SAR comparisons. This combined set of experimental and computational approaches is likely to be productive in uncovering additional mechanistic diversity within other families of structurally related molecules or chemical probes previously thought to be well characterized; a starting place for such studies might be families like indolocarbazoles and anthracyclines. Overall, the SAR-MAP platform establishes a systems-level approach to understanding the impact of specific chemical changes and has the potential to change how potency and selectivity are quantified in preclinical drug optimization.

## Methods

### PBMC collection and preservation
Peripheral blood mononuclear cells (PBMCs) were obtained with informed consent in accordance with the Declaration of Helsinki following protocols approved by Vanderbilt University Medical Center Institutional Review Board. All ethical regulations relevant to human research participants were followed. PBMCs were collected, isolated, and cryopreserved from approximately 20 mL of freshly drawn blood as previously described[66]. Briefly, peripheral blood was drawn into sodium heparin anticoagulant, and PBMCs were isolated by centrifugation after layering on top of a Ficoll-Paque PLUS (GE Healthcare Bio-Sciences) gradient.

### MV411 cell culture
MV411 is a cell line isolated from the blast cells of a 10-year-old male with biphenotypic B-myelomonocytic leukemia; they possess a FLT3-ITD mutation and translocation t(4;11)[57]. MV411 cells were obtained from ATCC (ATTC Cat#: CRL-9591, RRID:CVCL_0064) and confirmed to be mycoplasma negative. MV411 was cultured in IMDM (Gibco 12440-053), supplemented to a final concentration of 10% fetal bovine serum and 50 U mL$^{-1}$ of penicillin-streptomycin (HyClone, Thermo Fisher Scientific). Cells were incubated in a water-jacketed 5% $CO_2$ incubator at 37 °C and maintained at densities between 100,000 and 1

million cells per mL of culture media, fed every other day, and passaged every four days.

### eNSC collection and cell culture
All procedures involving animals were performed in accordance with animal health, safety, and wellness protocols outlined by both the institutional board (Institutional Animal Care and Use Committee) and national (National Institute of Health) governing bodies. All animal procedures were approved by the Vanderbilt University Institutional Animal Care and Use Committee (IACUC) review board (Vanderbilt IACUC protocol M2100037-00).

Timed pregnant mice (Wild-type C57 Black 6) were obtained from Charles River Laboratories (C57BL/6NCrl). eNSC Collection and Cell Culture followed the "Primary Cell Cultures" Methods section in Geben et al.[42]. Briefly, collection of mouse embryos was conducted at embryonic day 15.5 (E15.5) from timed pregnant dams. The collected cortical tissue from all embryos in each individual litter was then pooled, minced with microknives, and incubated at 37 °C, 5% $CO_2$ with 0.25% trypsin-EDTA solution for 20 min while rocking and then gently dissociated by trituration. Cells were cultured and maintained in embryonic neural stem cell specific media: Neurobasal media (ThermoFisher, 21103049); 1X B27 supplement without vitamin A (ThermoFisher, 12587010); 20 ng mL$^{-1}$ mouse epidermal growth factor (ThermoFisher, 53003018); 10 ng mL$^{-1}$ mouse basic fibroblast growth factor (ThermoFisher, PMG0035); 1 U/mL heparin (Sigma, 9041-08-01); 1X GlutaMax (ThermoFisher, 35050061); 1X modified Eagle's medium non-essential amino acids (11140050); 0.1 mM β-mercaptoethanol; 10 µg mL$^{-1}$ gentamicin[67]. Cells were fed every 2–3 days and passaged upon reaching confluence.

### H524 cell culture
H524 (human non-small-cell lung cancer cells) were obtained as part of U54 CA217450. H524 was cultured in RPMI (Thermo Fisher Scientific), supplemented to a final concentration of 10% fetal bovine serum and 50 U mL$^{-1}$ of penicillin-streptomycin (HyClone, Thermo Fisher Scientific). Cells were incubated in a water-jacketed 5% $CO_2$ incubator at 37 °C and maintained at densities between 200,000 and 800,000 cells per mL of culture media, fed every other day, and passaged every four days with Accutase dissociation.

### Curation of compound libraries
Initial screens were performed on a Diversity Set of 600 compounds selected from the BU-CMD's parent in-house small molecule screening collection that was comprised of 4445 compounds at the time of Diversity Set assembly (For representative prior screens conducted using the parent screening collection see refs. 68,69). The parent screening collection contained organic small molecules curated from Boston University synthetic organic chemistry laboratories working across a range of subdisciplines including diversity-oriented synthesis ("DOS"), medicinal chemistry, target oriented/natural products total synthesis, and synthetic methodology development. The Diversity Set was curated to contain a roughly proportional count of members from each of the 120 distinct structural chemotypes present in the parent screening collection. Of these, 23 chemotypes were flagged as "Natural Product-like", an in-house classification broadly encompassing small molecules that are either known natural products, natural product analogs, or synthetic intermediates generated enroute to the total synthesis of a natural product target. To assemble the Diversity Set, representative members of each chemotype (range: 1–23 compounds per chemotype; average: 5.4 members per chemotype) were randomly chosen from a preselected field of available parent library compounds meeting a specific stock volume threshold, to ensure availability of follow-up material for validation and secondary assays. The 600-compound Diversity Set contained 19 (3.1%) randomly selected members of the Natural Product-like "rocaglate" chemotype, and 9 (1.5%) additional randomly selected members of the biosynthetically related, Natural Product-like "aglain" chemotype.

### Fluorescent cell barcoding (FCB) assays

Fluorescent cell barcoding (FCB) is a multiplexing approach where cells within a given well are covalently labeled with different discrete levels of more than one $N$-hydroxysuccinimidyl ester functionalized amine reactive fluorescent dyes[9]. Using FCB, each well has a distinct fluorescent signature or barcode; this enables samples to be pooled for processing and running on the cytometer, thus reducing reagent consumption, and increasing throughput. The general protocol followed Schares et al. Barcoding plates were prepared as described in Section 6.1 of Schares et al.[70]. Briefly, the top level of Pacific Blue (PB) (level 8) was prepared at 10 µg mL$^{-1}$ in DMSO. PB level 7 was prepared by diluting 1 mL of PB level 8 into 710 µL of DMSO. Serial dilutions were repeated to produce 8 levels of PB. The top level of Pacific Orange (PO) (level 6) was prepared at 40 µg mL$^{-1}$ in DMSO. PO level 5 was prepared by diluting 1 mL of PO level 6 into 710 µL of DMSO. Serial dilutions were repeated to produce 6 total levels of PO. Ax750 was prepared at a concentration of 5 µg mL$^{-1}$ in DMSO and dispensed into the first half of a 96-well v-bottom polypropylene plate. 90 µL of PB was dispensed to rows A-H at decreasing concentrations and 90 µL of PO was dispensed to columns 1–6 of rows A-H at decreasing concentrations. See Supplementary Fig. 1 for schematic representation of the FCB technique. The plate was then centrifuged at $800 \times g$ for 5 minutes. 15 µL aliquots from the master plate were transferred into columns 1–6 and 7–12 of 96-well plates for later use, sealed with adhesive aluminum sealing film, and stored at −80 °C for up to 3 months.

### Diversity set experiment

For performing Diversity Set testing, the compound plate was prepared as follows: control compounds (three replicates of DMSO, staurosporine, etoposide, aphidicolin, rapamycin, and nocodazole) were prepared to final concentrations listed in Supplementary Table 3 and plated at 1 µL into columns 6 and 12 of eight 96 well tissue culture plates (Fisher). The Diversity Set compounds were acquired from the BU-CMD and plated at a volume of 1 µL into the remaining 80 wells of the eight 96-well tissue culture plate for a final concentration of 10 µM.

MV411 cells from suspension culture (0.5-1 million cells mL$^{-1}$) were acquired, pelleted, and resuspended in media to around 500,000 cells mL$^{-1}$ at least 2 hours before start of the experiment as described in Schares et al. Cell suspension was dispensed at 200 µL to each of the eight 96 wells plates and pipetted to mix. The plate was incubated at 37 °C, 5% CO$_2$ for 16 hours. The remaining steps were performed as described in Schares et al. Briefly, cells were stained for viability with 0.04 µg mL$^{-1}$ Alexa 700 SE (Ax700-SE), fixed with 1.6% paraformaldehyde, and permeabilized with 100% ice-cold methanol (for more information on the use of Ax700-SE for viability testing, see Table 2 in ref. [7]). Cells were stained using eight concentrations of Pacific Blue and six concentrations of Pacific Orange (per 48 wells) for generating a unique fluorescent barcode for each well, and one concentration of Alexa Fluor 750 as an internal control. Cells were pooled into one flow cytometry tube per 48-well plate (16 tubes total) for staining with the following antibody panel: c-CAS3, γH2AX, p-S6 S240/244, and p-HH3 (more information detailed in Supplementary Table 2). Compensation controls for each antibody and dye and bead controls were prepared and used for the set-up of the flow cytometer. Flow cytometry data was acquired on a 5-laser (355 nM, 405 nM, 488 nM, 561 nM, and 635 nM) BD LSR II Fortessa instrument.

### Rocaglate set experiment

For the Rocaglate Set experiment, the compound plate was prepared as follows. Control compounds (3x DMSO, staurosporine, etoposide, **CMLD010335**, rapamycin, and nocodazole) were prepared to final concentrations listed in Supplementary Table 3 and plated at 1 µL in columns 6 and 12 of a 96 well tissue culture plate. The Rocaglate Set (RR = 20, ADR = 8, RP = 9) acquired from the BU-CMD was plated at a volume of 1 µL into the remaining 80 wells of the 96-well tissue culture plate at a final concentration of 10 µM.

As described above, MV411 cells from suspension culture were acquired, pelleted, and resuspended in media at least 2 hours before the start of the experiment. PBMCs were thawed from cryopreservation and resuspended at ~$2 \times 10^6$ cells mL$^{-1}$. H524 from culture was treated with Accutase (ThermoFisher) for 5 min to dissociate cells adhering to flask, pelleted and resuspended at ~$2 \times 10^6$ cells mL$^{-1}$; they also required Accutase dissociation at the end of the 16-h time points. eNSC from culture was treated with both Accutase and 10 µM ROCK inhibitor (Y-27632 dihydrochloride, Bio-Techne) for 5 min, pelleted, and resuspended at ~$2 \times 10^6$ cells mL$^{-1}$ at the beginning of the 16 hour time points; they were also treated with Accutase and ROCK inhibitor at the end of the 16 hour timepoint. Cell suspensions were dispensed at 200 µL to each of the 96 wells of 4 plates, 48 wells for each cell type, and pipetted to mix. The plate was incubated at 37 °C, 5% CO$_2$ for 16 hours. The remaining steps were performed as described above, pooling cells into two total flow cytometry tubes for antibody staining: one for PBMC and one for MV411[70]. Antibody staining panel included p-STAT3, p-STAT5, p-ERK, p-HH3, p-4EBP1, p-S6 S240/244, p-S6 S235/236, Ki67, and γH2AX (Supplementary Table 2). Flow cytometry controls were performed as above. Flow cytometry data was acquired on a four laser (405 nM, 488 nM, 561 nM, and 640 nM) Cytek Biosciences Aurora spectral flow cytometer following spectral unmixing with compensation controls.

### Dose-response

**CMLD012390, CMLD013342**, and **CMLD013608** were prepared at the following doses and plated at a volume of 1 µL into the first two columns of rows A-H of a 96-well plate, respectively: 0.01, 0.05, 0.1, 0.5, 1, 5, 10, and 0 µM.

As described above, MV411 cells from suspension culture were acquired, pelleted, and resuspended in media at least 2 hours before the start of the experiment. The cell suspension was dispensed at 200 µL to each of the two columns of the 96 well plate and pipetted to mix. The plate was incubated at 37 °C, 5% CO$_2$ for 16 hours. Cells were stained for viability, fixed, and permeabilized. Cells for each compound were then stained using eight concentrations of Pacific Blue, one concentration of Pacific Orange, and one concentration of Alexa Fluor 750. Cells for each compound were pooled into their own flow cytometry tube for staining with the same antibody panel as the Rocaglate Set experiment (Supplementary Table 2). Flow cytometry controls and data collection were performed as in the Rocaglate Set experiment.

### 4'-Methoxy B-Ring SAR analysis

DMSO, etoposide, **CMLD012390, CMLD013342, BUCMD00617**, and **BUCMD00616** were plated at a volume of 1 µL into the first row of columns 1–6 of a 96-well plate. All compounds were tested at 10 µM.

As described above, MV411 cells from suspension culture were acquired, pelleted, and resuspended in media at least 2 hours before the start of the experiment. The cell suspension was dispensed at 200 µL to each of the six columns of the first row of the 96 well plate and pipetted to mix. The plate was incubated at 37 °C, 5% CO$_2$ for 16 hours. Cells were stained for viability, fixed, and permeabilized. Cells for each compound were then stained using four concentrations of Pacific Blue, one concentration of Pacific Orange, and two concentrations of Alexa Fluor 750. Cells for each compound were pooled into their own flow cytometry tube for staining. The antibody staining panel included c-CAS3, p-HH3, p-4EBP1, p-S6 S240/244, Ki67, and γH2AX (Supplementary Table 2). Flow cytometry controls and data collection were performed as in the Rocaglate Set experiment.

### FCB data preprocessing and analysis

Data was uploaded and stored in Cytobank for scaling, quality control gating, compensation, and analysis of unmixed cytometry data (FCS file format). Raw median fluorescence intensity values were transformed to the inverse hyperbolic sine (arcsinh) scale. For the Diversity Set test, a cofactor of 150 was selected for all functional readouts. For the Rocaglate Set experiment, the default cofactor of 6000 was selected for all markers except

for Ki67 (25000), p-S6 S240/244 (12000), p-LCK (12000), p-STAT3 (12000), p-STAT5 (25000), p-S6 S235/236 (12000), p-HH3 (12000), and p-4EBP1 (12000). For the dose-response and SAR experiment, a cofactor of 6000 was selected for all functional readouts. Quality control gating (QC) was performed as shown in Supplementary Fig. 10. Scaled and gated samples were then compensated. Diversity Set and Rocaglate Set experiments were computationally deconvoluted using the DebarcodeR algorithm and the resulting FCS files for each well were uploaded to Cytobank for storage and further analysis. The dose response and 4'-methoxy B-ring SAR experiments were manually deconvoluted directly in Cytobank.

### Dimensionality reduction analyses

The dimensionality reduction tool used here was t-SNE-CUDA through Cytobank. All t-SNE analyses were performed as follows: after debarcoding, the FCS files corresponding to stained cellular events from all wells were uploaded to one Cytobank experiment and fed into a t-SNE-CUDA analysis through Cytobank (settings: channels = 11 functional readouts included in the panel, iterations = 10,000, perplexity = 60, automatic learning rate, early exaggeration = 12). FCS files containing protein measurements for all readouts with the t-SNE axes appended were exported into R for subsequent analysis.

### T-REX analyses

T-REX takes a pair of dimensionally reduced maps of equal size as inputs, creates a plot that highlights hotspots of cells in phenotypic regions that are the most different between the two files, and provides a T-REX degree of difference value for each analysis performed. Following dimensionality reduction as described above, each map was sampled such that each well and rocaglate structural subclass, when applicable, was equally represented before applying T-REX. The T-REX algorithm was applied in R using a k value of 60. A modular data analysis workflow including UMAP as the dimensionality reduction tool, K-Nearest Neighbors (KNN), and Marker Enrichment Modeling (MEM) was developed in R and is available online (https://github.com/cytolab/T-REX).

### Hierarchical clustering analysis

The log-like arcsinh ratio of the median phospho-protein signal in treated cells as compared to the signal in the first vehicle was performed for each marker and compound in the Rocaglate Set in Cytobank. FCS files containing transformed median fluorescence intensity values for the 37 rocaglates and 3 vehicles were then imported into R. The hclust function from the stats package was used with the default settings to perform complete linkage hierarchical clustering on the transformed median fluorescence intensity values. The top five clusters from the hierarchical clustering analysis were displayed.

### MEM protein enrichment analyses

Marker Enrichment Modeling from the cytoMEM package available on Bioconductor (https://www.bioconductor.org/packages/release/bioc/html/cytoMEM.html) was used to characterize feature enrichment in the RP island identified. MEM has two ways of defining a reference point. In the original, a chosen set of cells, such as a control or "all other cells", was used to calculate a comparative MEM label[36]. For KNN-based uses of MEM, a statistical reference point was defined based on the full dataset[35]. Here, a statistical reference point intended as a statistical null hypothesis was used as the MEM reference. For this statistical null MEM reference, the magnitude was zero and the IQR was the median IQR of all features chosen for the MEM analysis. Values were mapped from 0 enrichment to a maximum of +10 relative enrichment. The contribution of IQR was zeroed out for populations with a magnitude of 0.

### Cell gating analyses

Gating was performed in Cytobank to quantify the % γH2AX positive cells in response to each compound in MV411 and PBMC. This analysis was performed by plotting Side Scatter Area vs. γH2AX for each compound and cell type and using the polygon gate tool to draw an area that maximizes the difference in percentage % γH2AX positive cells between the vehicle and control wells. Quantifying the percentage of cells with the MADD signaling profile was also performed using the polygon gate in Cytobank.

### Velociraptor-eye

Velociraptor-Eye was developed in Cross et al.[37]. The MEM phenotype used as the search label (Fig. 5c) was the output of MEM protein enrichment analysis (cytoMEM in Bioconductor) of the indicated cells with the MADD signature (Fig. 5c). Following dimensionality reduction of the dose response dataset as described above, k-nearest neighbors (using k = 60 cells) was run on every cell in the embedding using the fast nearest neighbors package in R. The phenotype of each cell neighborhood was then quantified by MEM. The similarity of each neighborhood to the search MEM label was calculated as root-mean-square deviation.

### EC$_{50}$ calculations

EC$_{50}$s were calculated in R using the drm function from the drc package. A four-parameter log-logistic function was selected with the following unfixed parameters: hill slope, minimum value, maximum value, and EC$_{50}$. The dependent variable for the dose-response formula was the percentage of cells per condition with greater than 75% similarity to the search label identified by Velociraptor-Eye. The independent variable was the common logarithm of the dose for each concentration.

### Statistics and reproducibility

For all statistical analyses, methods are described in the relevant part of the main text, figure legends, and/or Methods section. Significance threshold for all statistical analyses in paper was α = 0.05. Experiments were not randomized and the sample sizes, including number of compounds and live cells used for analysis, are listed within the figures, main text, figure legends and/or Methods section. Statistical methods were not utilized for determining sample sizes. Measurements for the Diversity Set, Rocaglate Set, and dose response data sets were taken from distinct samples.

### Reporting summary

Further information on research design is available in the Nature Portfolio Reporting Summary linked to this article.

### Data availability

The FCS files used for generating all figures in the main manuscript and supplemental information (including for the Diversity Set, Rocaglate Set, Dose Response, and 4'-methoxy SAR data sets) are deposited at Zenodo and publicly available at the following URL: https://doi.org/10.5281/zenodo.15530850. All other data are available from the corresponding author (or other sources, as applicable) on reasonable request.

### Code availability

Descriptions for conducting all analyses, including specific R packages, functions, and parameters can be found in the relevant Methods section above and all code used is publicly available (links in methods section). Custom code was not developed for this manuscript.

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

## Acknowledgements

Research was supported by the following funding resources: NIH/NCI grants U01 TR002625 (H.L.T., M.J.H., L.E.B., J.A.P., and Jr., J.M.I.), R01 CA226833 (J.M.I., H.L.T., and M.J.H.), R01 NS118580 (R.A.I. and L.C.G.), R35 GM118173 (J.A.P., Jr.), T32 GM149371 (H.L.T.), T32 GM007628 (L.C.G.), and F31 NS120608 (L.C.G.). The authors would like to thank Claire E. Cross for her assistance in performing the Velociraptor-Eye analysis.

## Author contributions

All authors contributed to developing the study and reviewing and editing the manuscript. L.E.B. and J.A.P., Jr. developed the process to select molecules for the Diversity Set and Rocaglate Set, provided access to the Diversity Set and Rocaglate Set, and contributed to rocaglate subtype binning, R-group decomposition, and structure-activity analysis. L.C.G. and R.A.I. assisted with generation and interpretation of data and writing. M.J.H. and H.L.T. collected and processed data. H.L.T. and J.M.I. conducted flow cytometry data analysis and interpretation. H.L.T. developed data analysis scripts. H.L.T. and J.M.I. drafted the manuscript. J.M.I. and J.A.P., Jr. provided financial support.

## Competing interests

The authors declare no competing interests.
