## [Transparent Peer Review file · Communications Biology]

Connecting Chemical Structure to Single Cell Signaling Profiles

Corresponding Author: Dr Jonathan Irish

Version 0:

Reviewer comments:

Reviewer #1

(Remarks to the Author)

In this study, Thirman et al screen 600 molecules for their bioactivity in the human cell line MV411. Phospho-flow is used as read-out, with emphasis on four marker proteins. Sixty-five molecules were identified as active, and of these rocaglates constituted the largest chemotype. These are therefore characterized in further detail. The main aim of the study was to investigate structure-activity relationships, but the authors also study the specificity of the molecules towards leukemia cells. A limitation with this part of the study is that the authors use a leukemia cell line (MV411) as their positive control, while primary PBMCs from a healthy donor is used as a negative control. It is not clear why the MV411 cell line was chosen, and why the authors did not use primary leukemia cells which would also include internal healthy cells as a control in the same sample. Furthermore, the authors mention in the abstract that their method can be applied to animal models, but this is not demonstrated here.

Overall, the study is well presented and the figures are beautiful. I have some suggestions for improvements as described below:

1. In the abstract, it would be helpful if the assay used for testing the 600 compounds could be mentioned. I.e. in the sentence starting with "Testing >600 representative molecules identified rocaglates...". It would also be of interest to include the number of rocaglates identified in the same sentence.
2. Perhaps the authors should reconsider the reference to animal models in the abstract since this was not studied here. The possibility of applying the method to animal models could be mentioned in the Discussion instead.
3. The authors include a significant description of their study design and results in the Introduction, even referencing to several figures. I would recommend moving the results to the Results section and only briefly summarize them in the Introduction.
4. When referring to Figure 1 (page 4), it would be helpful to also include what panel the reader should look at.
5. Overall, the figures are beautiful and the color codes are consistent. One limitation is the use of red and green, as these colors cannot be distinguished by the color blind. This is of particular importance in Figure 3b, which cannot be appreciated fully by this group. I would recommend choosing a different color scheme.
6. If I understand correctly, the screen was performed once in the MV411 cell line (i.e. not repeated). Could the authors include the rationale for this, and discuss possible limitations of not repeating the screen?
7. In the legend to Figure 3, I believe the reference to Figure 2b should be to Figure 3a?
8. In Figure 4a, color codes are used for the different activity profiles of the compounds (left). Could the authors please include a reference to these colors in the main text when describing the compound groups? This would make it easier to refer to the figure. Please also add the reference to the figure after the sentence "Only the vehicle and three RRs...". It would be helpful to be presented here what RRs the authors are referring to.

9. The authors identify a subpopulation of MV411 cells that show a MADD signaling profile. Can the authors please discuss how come there is a subpopulation in a cell line, which is supposed to be homogeneous? It would be more intuitive to understand this pattern if it was present in PBMCs.
10. On page 15, there is a spelling mistake: "proteins on interest" should be "proteins of interest".
11. In methods, can the authors please include information on what type of leukemia cells the MV411 cells are, and what was the rationale for using this model?
12. On page 17, is it correct that "Ax700" was used for viability stain? Something is missing?
13. The authors write both "arcsine" and "arcsinh" (figures and text). Are these the same thing?

Reviewer #2

(Remarks to the Author)

This article, titled "Connecting Chemical Structure to Single Cell Signaling Profiles" by Thirman et al., presents an approach called SAR-MAP (Structure-Activity Relationship using Multiplexed Activity Profiling) for high throughput ex vivo characterization of cellular activities in response to multiple chemical structures. A strength of this work is the use of fluorescent cell barcoding to measure multiple cellular readouts simultaneously at the single-cell level (note that this method was published previously). The method is interesting and has the potential to be applied to study other cells and molecules.

1. More details are needed re the FCB procedure. The labeling steps are included in the method, but it is not entirely clear to this reviewer what cells are multiplexed. Each well on the plate is for a different compound at a different concentration, and then these wells were barcoded/mixed? Some type of schematic representation would go a long way for readers not familiar with this technique.
2. The heterogeneity within MV4;11 is intriguing but also raises questions. This is a cancer cell line which is clonal at least at the DNA genomics level. What is driving the variation in drug response? How do we know these are not just stochastic? It would be a lot more convincing if this is done using PDX-derived cells which often consist of multiple clones etc.
3. As only one cancer cell line and one control are studied, the validity of the biological discoveries in terms of the activity and selectivity of the prioritized compound may need further studies to confirm. I suggest the authors mention this in the discussion. Or they could consider mixing two different cell lines each with a differential drug response profile.
5. PBMC includes a multitude of cell types and these experiments (I assume for safety/toxicity signals) are cursory and uninformative
4. The EC50 calculation is not clear to this reviewer. Since similarity with the MADD score can be calculated for each cell, why is clustering (by DBSCAN) necessary and why only selected clusters are used? There is a steep change in MADD signaling between 0.1 to 0.5 μ M for the two RPs, resulting in EC50 of ~600nM. This seems to suggest that the effective concentration ranges of RPs to activate MADD are narrow. However, the concentration used for profiling was 10nM, which is much lower than the concentration needed to activate MADD in the two RPs. Can the authors explain this discrepancy?

Reviewer #3

(Remarks to the Author)

Thirman et al. performed a diversity library screen by flow cytometry, identifying Rocaglates as a chemical class with specific activity on leukemic cells. Whilst the pharmacology on leukemia cells has been previously demonstrated, the authors go on to leverage multiparametric flow cytometry in order to characterize sub-classes of molecules exhibiting differentiated pharmacology, and relate these back to structural features of the molecules.

Overall, certain claims are made around novelty that I'm unsure of. For example, Rocaglates anti-leukemic activity is well established (Callahan et al. - <https://ashpublications.org/blood/article/120/21/1338/89568/Rocaglamide-Selectively-Eradicates-Human-Leukemia>). Several other papers have used multi-parametric flow cytometry endpoints (e.g. see Ding et al. - <https://pubmed.ncbi.nlm.nih.gov/30807094/>; Tuijnenberg et al - <https://pubmed.ncbi.nlm.nih.gov/31621069/>). The integration with multiparametric visualization is a slight advance here, however the data is used in a more qualitative way to describe and bin compound subclasses. Finally, claims about the utility of these approaches as they relate to drug discovery are a stretch, given that it's unclear to me why multiplexing the endpoints in the same assay is advantageous over simply running several, discrete, medium throughput assays. However to this final point, authors could perhaps discuss the importance of generating multiparametric data for natural product/phenotypic screening libraries early on in a drug discovery campaign versus sequential generation of this data.

What was most interesting was the authors delineation of subclasses of Rocaglates, which were previously considered to be a single class. In particular, the MADD signature, represented in a small fraction of cells (but potentially indicative of a broader activity in the entire population) significantly highlights the benefits of integrating single-cell techniques into early SAR studies. This for me is the key advance in the paper, that could be elaborated and emphasized more in the conclusions.

Major:

1) Some of the pharmacology terminology should be improved, to avoid confusion with more standard use. I would recommend replacing 'bioactivity' with 'pharmacodynamic effects', as it is referring to a specific phenotypic readout (gH2ax

induction). 'Selectivity' tends to be used to refer to specific molecular interactions, and should be replaced with 'selective activity'. 'target activity' should be referred to as 'proximal target engagement'. This should be amended throughout the manuscript and figures.

2) Related to 1, some of the terminology is inconsistent throughout the manuscript. pg.6 paragraph 1 is a specific example of this: "Exceptional activity following treatment with a compound can be measured in different ways, including exceptional target activity or potency (i.e., clamping eIF4A1 at a given concentration, half maximal inhibitory concentration)". This statement is simply referring to potency and efficacy parameters for a measure of proximal target engagement. The authors should take a careful look at the pharmacology terms throughout the paper to ensure consistency and clarity.

3) The 5th paragraph in the introduction should be removed, and integrated with the first paragraph of the results section.

4) Throughout the manuscript, data is all generated and interpreted at a single (high) concentration. Whilst this approach makes sense for selection of initial compounds, it is very challenging to interpret the rest of the manuscript without an understanding of the dose response. The way I might resolve this is to explore dose responses for some key compounds and major conclusions from the paper. For example – in figure 3A, it would be helpful to understand whether some of the less active compounds are simply a result of where they sit on their dose response curve versus more active compounds - i.e. is the SAR observed related to potency, efficacy or distinct pharmacology?

5) It would greatly add to the paper to perform basic cell killing assays (beyond simply gH2AX or cleaved caspase), and to show differential pharmacology of the MADD compounds versus non-MADD profile compounds. For example, demonstrating a differential between proximal target engagement potency and cell killing that correlates to the MADD characteristics would show biological significance of this finding. As it stands the results are fine but somewhat descriptive, whereas linking the subclasses of compounds back to a biologically meaningful phenotype would be a powerful demonstration of approach taken.

6) The advantages of multiplexing assays (e.g. gH2ax, phosphor endpoints), versus more traditional and higher throughput approaches such as immunofluorescence should at a minimum be discussed. Typically in a drug discovery setting, running multiple assays in medium throughput is not rate limiting, and its unclear whether the sensitivity of the flow cytometry assay is comparable to other more established methods such as gH2AX imaging. This being said, generating multiparametric data on initial compounds rather than sequentially can certainly have advantages to choose starting matter or accelerate med chem efforts (vs finding out later into a med chem campaign).

Minor:

7) More information should be provided on the 'diversity set' of compounds used for initial profiling. E.g. was this a commercial or proprietary compound library, were compounds all natural product derived, are their previous publications characterizing the library? This can help to better contextualize the hits observed in figure 1.

8) A representative of the amidino-rocaglate structural subclass should also be shown in supplementary figure 1 to better enable comparison of structure

9) The introduction should also include mention/citation of the mechanism of action of rocaglates that underpins their selective activity in leukemia cells – e.g. if the differential effects on leukemia vs normal is due to the high growth/proliferation rates of leukemic cells.

10) Pg. 6 paragraph 1. "Exceptional activity following treatment with a compound can be measured in different ways, including exceptional target activity or potency (i.e., clamping eIF4A1 at a given concentration, half maximal inhibitory concentration.." It is very challenging to make claims about exceptional activity based on activity 'at a given concentration' . Rather, the activity should be based on potency or efficacy of a compound. This should be clarified in the text

11) Figure 4C - is it understood why the side scatter parameter is so variable between the examples shown? Was SSc correlated with any other parameters across compounds?

12) Pg.9 gH2AX induction can be a function of compound potency, efficacy and time. In addition, it is not clear why a 15% cut off is used to determine if compounds trigger DNA damage. Overall this is a fairly qualitative way to evaluate compounds. The gating strategy (Fig3C) is also questionable, in that very small shifts in MFI are having a large impact on the % positive due to the tight gating. An orthogonal assay (e.g. gH2AX immunofluorescence assay) would build confidence in the authors conclusions.

13) The dose response data in Supplementary Figure 7 is very nicely supporting of the qualitatively distinct SAR, and could be incorporated into the main text, as part of Figure 5.

14) Authors could consider citing the following review articles, or integrating some of their cited references: Ullas et al. (<https://pubmed.ncbi.nlm.nih.gov/38612661/>), which has a section on some of the ways flow cytometry has been applied to early phenotypic screening; Vincent et al. (<https://pubmed.ncbi.nlm.nih.gov/35637317/>), which highlights the limitations of single-parameter phenotypic screening, and includes discussion of applying multiparametric endpoints.

Version 1:

Reviewer comments:

Reviewer #1

(Remarks to the Author)

The authors have addressed all my comments in a clear and structured way. I have no additional comments and congratulate the authors on the nice work!

Reviewer #2

(Remarks to the Author)

Thanks for carefully addressing my comments.

Reviewer #3

(Remarks to the Author)

I want to thank the authors for integrating feedback from my comments, and from the other reviewers. I think the paper is much stronger with additional data that solidifies the key finding and have no further questions.

This previously submitted manuscript underwent major revision to address all reviewer points. We believe these changes have resulted in a substantially improved manuscript. Here is a summary of the key points and changes made in response, followed by individual reviewer critiques (black text) and author responses (blue text).

Major points raised collectively by the editors and reviewers were:

- 1) Reviewers requested data in the main text showing dose responses for key compounds; these data were added to Figure 6d and a schematic in Figure 6c (details below).
- 2) Reviewers requested basic cell killing data from the experiments quantifying the MADD signaling profile; these data were added to Figures 3b & 3d (details below). They also requested clarification of terminology and alignment with field standards in describing results, especially from Figure 3; these changes were made with Figure 3 and throughout the Results text.
- 3) Reviewers requested tests of rocaglates in additional cell types to address reproducibility of results and provide the rationale for the selection of MV411; the results of these tests and the rationale are described in the Results text and the associated new data are shown in Supplementary Figure 3.

Addressing #1, dose responses for key compounds

- Dose response data for two key compounds – CMLD012390 and CMLD013342 - including two EC50 curves were moved from the Supplementary Information into Figure 6d.
- A new schematic was added as part of Figure 6c to clarify the methods used to calculate EC50. This addressed a reviewer point that dose response information validating the activity for these two key compounds should be more prominently emphasized in the main text.

Addressing #2, basic cell killing data

- The fold change in cell count for treated wells as compared to vehicle cell count was calculated (i.e. cell count treated / cell count vehicle) and added to Figure 3b and Figure 3d. Given that this dataset was gated for live, intact, single cells as part of pre-processing, cell count acts as a measure of basic cell killing.
- The addition of cell count data to Figure 3 correlated the results associated with cell killing seen across rocaglate structural subclasses presented in Figure 3a and dendrogram organized clusters presented in Figure 3c with a measure of cell death. These findings added substantial text to the Results and Discussion and clarified use of terms like bioactivity and pharmacodynamic effect.

Addressing #3, testing in additional cell types, rationalizing MV411, and reproducibility

- Data and analysis for two additional cell types – H524 (lung cancer cell line) and eNSC (healthy primary mouse embryonic neural stem cells) – were added as part of Supplementary Figure 3 and described in new text in the Results and Discussion, addressing two key reviewer points:
 1. A request to see that rocaglates are bioactive in additional cell types and that key results hold true in other cell types (e.g., γ H2AX activation was observed in all cell types, Supplementary Figure 3).
 2. MV411 and PBMC were selected on the basis of subclass specific activity being observed, higher cell counts, and lower variation within the vehicle data.
- Additional text was added to the Discussion addressing rigor and quantifying reproducibility; these results quantify the high level of reproducibility of the SAR-MAP approach; specific changes:
 - o Text and analysis quantifying the reproducibility of the two focal RPs (CMLD012390, CMLD013342) tested in three independent repeat experiments.

- Text and a figure quantifying the low level of variation in vehicle controls in MV411 and PBMC (Supplemental Figure 3), which improves sensitivity to detect rocaglate effects on cells.
- Text was added to the Discussion regarding rigor and reproducibility in this study, exploring limitations of the current study, and highlighting ways to potentially expand and improve the approach.
- Noted throughout the text and figures the number of cells measured.

Specific point by point responses:

Reviewer #1 (Remarks to the Author):

In this study, Thirman et al screen 600 molecules for their bioactivity in the human cell line MV411. Phospho-flow is used as read-out, with emphasis on four marker proteins. Sixty-five molecules were identified as active, and of these rocaglates constituted the largest chemotype. These are therefore characterized in further detail. The main aim of the study was to investigate structure-activity relationships, but the authors also study the specificity of the molecules towards leukemia cells. A limitation with this part of the study is that the authors use a leukemia cell line (MV411) as their positive control, while primary PBMCs from a healthy donor is used as a negative control. It is not clear why the MV411 cell line was chosen, and why the authors did not use primary leukemia cells which would also include internal healthy cells as a control in the same sample.

We appreciate this point. In response, we have added data from two additional cell types (Supplementary Figure 3) that helps explain the decision to carry forward SAR-MAP analysis in MV411 and PBMC (these were the cell types with the lowest variation in the vehicle controls across all tested readouts, which helps to make the assay more sensitive and reproducible). We also included new text in the Discussion explaining plans to test top molecules in primary leukemia samples in future studies.

Discussion Text:

Lines 411-413: “Due to the inherent variability of cells, even within a clonal population, future studies should include replication of the Rocaglate Set screen on MV411, other leukemia cell lines, and primary leukemia samples to validate the contrasting bioactivity and signature profiles across structural subclass.”

Lines 418-420: “Thus, future experiments should include a time course and testing on primary leukemia with cell surface markers to resolve the cellular mechanisms and subpopulations driving the MADD signaling profile.”

Supplementary Figure 3 – Rocglates demonstrate contrasting bioactivity across subclass in MV411 and PBMC in comparison with H524 and eNSC. Heatmaps depicting the arcsinh ratio of the median fluorescence intensity for each compound (listed on top) and readout (listed on left) by the median fluorescence intensity of Vehicle 3 for MV411, H524, eNSC, and PBMC, respectively. Cells on the heatmap range from light blue for the lowest values to bright yellow for the highest values. Compounds are grouped and colored according to the rocglate subclass listed on the top of the plot.

Furthermore, the authors mention in the abstract that their method can be applied to animal models, but this is not demonstrated here.

We appreciate this point from the reviewer and have removed the reference to animal models in the abstract.

Overall, the study is well presented and the figures are beautiful. I have some suggestions for improvements as described below:

1. In the abstract, it would be helpful if the assay used for testing the 600 compounds could be mentioned. I.e. in the sentence starting with "Testing >600 representative molecules identified rocaglates...". It would also be of interest to include the number of rocaglates identified in the same sentence.

We appreciate this point and have updated the Abstract text to include both points: 1) text describing the assay and 2) the number of rocaglates identified.

Lines 18-19: "Testing 600 representative molecules using MAP revealed roughly half of tested rocaglates (9 of 19) were bioactive."

2. Perhaps the authors should reconsider the reference to animal models in the abstract since this was not studied here. The possibility of applying the method to animal models could be mentioned in the Discussion instead.

As described above, we have removed the reference to animal models in the abstract.

3. The authors include a significant description of their study design and results in the Introduction, even referencing to several figures. I would recommend moving the results to the Results section and only briefly summarize them in the Introduction.

We thank the reviewer for this comment; to address this, we removed the study design description from the Introduction paragraph and moved these points to the Results (1st, 3rd, and 4th paragraphs of Results).

4. When referring to Figure 1 (page 4), it would be helpful to also include what panel the reader should look at.

We agree and have added references to Figure panels whenever the associated results are from a single panel (in this case, clarifying that the associated panel is Figure 1d, line 112).

5. Overall, the figures are beautiful and the color codes are consistent. One limitation is the use of red and green, as these colors cannot be distinguished by the color blind. This is of particular importance in Figure 3b, which cannot be appreciated fully by this group. I would recommend choosing a different color scheme.

We thank the reviewer for this comment and appreciate their concern about figure colors and design choices being more accessible. We have modified Figure 3a to include symbols associated with each rocaglate structural subclass. These symbols are used in Figure 3c such that rocaglate structural subclass membership can be identified without the need to distinguish red and green. The remaining figures have text associated with each rocaglate structural subclass, such that the colors are not necessary for interpretation of results.

Figure 3 – Rocaglate subclasses had distinct patterns of bioactivity. **a)** Heatmap depicting the arcsinh ratio of the median fluorescence intensity for each compound (listed on top of heatmap) and readout (listed left of heatmap) by the median fluorescence intensity of Vehicle 1. Cells on the heatmap range from light blue for the lowest values to bright yellow for the highest values. Compounds are grouped and colored according to the rocaglate subclass listed on the top of the plot. **b)** Heatmap of the cell count for each compound divided by the cell count for Vehicle 1. Cells on this heatmap range in color from black, for molecules with low cell count relative to vehicle, to yellow for molecules with a similar cell count to vehicle. Heatmap is clustered according to rocaglate structural subclass as displayed in **Figure 3a**. **c)** Heatmap as in **Figure 3a** clustered according to a dendrogram of the transformed median fluorescence intensity for each compound and readout. **d)** Heatmap as in **Figure 3b** clustered according to a dendrogram displayed in **Figure 3c**.

6. If I understand correctly, the screen was performed once in the MV411 cell line (i.e. not repeated). Could the authors include the rationale for this, and discuss possible limitations of not repeating the screen?

We appreciate this point and realize we should highlight better in the text replicates (e.g., all vehicle is done in triplicate, now explicitly written on line 144) and experimental repeats (e.g., all key compound findings were repeated in three independent experiments). In addition, we added a new figure and new data from

two additional cell types, one lung cancer cell line and one healthy murine primary cell, as part of Supplementary Figure 3 on page 3 above (H524 and eNSCs, respectively). These new data complement the leukemia cell line data and healthy human primary cell data (MV411 and PBMCs, respectively). The associated text describing this figure can be found on lines 144-162.

Text changes regarding rigor and reproducibility included text in the Discussion regarding the reproducibility of the key results from the top two key RPs (CMLD012390 and CMLD013342) which have been observed in at least three separate experimental repeats (lines 415-417). Additional text on rigor, reproducibility, and limitations was added, especially in the Discussion (lines 410-414).

Lines 414-416: “The prominent result of cooccurring γ H2AX, p-4EBP1, and p-S6 S240/244 activity from the top two molecules identified - CMLD012390 and CMLD013342 - replicated in the two additional studies presented, thus providing encouraging evidence that the SAR-MAP platform is reliable and reproducible.”

Lines 409-413: “While the Diversity Set and Rocaglate Set screens were only performed once in each cell line, the single cell resolution of phospho-flow data offers some built-in replication in comparison with a platform solely detecting bulk cellular responses. Due to the inherent variability of cells, even within a clonal population, future studies should include replication of the Rocaglate Set screen on MV411, other leukemia cell lines, and primary leukemia samples to validate the contrasting bioactivity and signature profiles across structural subclass.”

7. In the legend to Figure 3, I believe the reference to Figure 2b should be to Figure 3a?

Yes, thank you -- we have corrected the reference to say Figure 3a.

8. In Figure 4a, color codes are used for the different activity profiles of the compounds (left). Could the authors please include a reference to these colors in the main text when describing the compound groups? This would make it easier to refer to the figure. Please also add the reference to the figure after the sentence “Only the vehicle and three RRs...”. It would be helpful to be presented here what RRs the authors are referring to.

We appreciate this point from the reviewer and have made these changes. We added text clarifying the color of bars being referred to when discussing data related to each rocaglate structural subclass, including specifically a reference to Figure 4a after the sentence “Only the vehicle and three RRs...” to clarify which specific RR compounds are being referenced.

Lines 223-229: Only the vehicles (purple) and three RRs (red) did not trigger significant DNA damage in either cell type, further demonstrating the exceptional bioactivity of rocaglates (Figure 4a). Nearly all ADRs (orange) and one RR (green) (SDS-1-021 43, in the set as both racemic (CMLD011880) and enantioenriched (CMLD010508) stocks) triggered significant DNA damage in PBMC alone; these rocaglates displayed selective activity for targeting healthy blood, again showcasing the power of the SAR-MAP platform to reveal diverse effects, including those desired and those not. All RPs (green) and some RRs (red) led to the desired leukemia-selective induction of γ H2AX.

9. The authors identify a subpopulation of MV411 cells that show a MADD signaling profile. Can the authors please discuss how come there is a subpopulation in a cell line, which is supposed to be homogeneous? It would be more intuitive to understand this pattern if it was present in PBMCs.

We thank the reviewer for this comment and discuss how differences observed in some cells within clonal cell lines can be due to non-genetic effects (e.g., cell cycle stage, stochastic variation in epigenetic and protein states within cells). These differences are routinely observed cell lines when using single cell approaches like phospho-flow or even simple analysis of surface protein expression and are present in the cell lines immediately after ordering from ATCC or clonal derivation. We have text to the Discussion (lines

369-381) noting these points.

10. On page 15, there is a spelling mistake: “proteins on interest” should be “proteins of interest”.

We have fixed this noted spelling error.

11. In methods, can the authors please include information on what type of leukemia cells the MV411 cells are, and what was the rationale for using this model?

We appreciate this point from the reviewer and have included further information on the patient and cytogenetics of MV411 leukemia cells in the Methods subsection title “MV411 Cell Culture” (lines 439-440). Additionally, as described above, we have added supporting data (Supplementary Figure 3) and text to the Results (lines 144-162) on the rationale for using MV411 for the initial Diversity Set screen and subsequent SAR-MAP analysis (briefly, these cells had the most reproducible baseline signaling and responses to compounds across replicates and experimental repeats).

Lines 439-440: “MV411 is a cell line isolated from the blast cells of a 10-year old male with biphenotypic B-myelomonocytic leukemia; they possess a FLT3-ITD mutation and translocation t(4;11)”

12. On page 17, is it correct that “Ax700” was used for viability stain? Something is missing?

We thank the reviewer for this question about the notation ‘Ax700’, which indicates the channel on which we measured the per-cell accumulation of Alexa Fluor 700 succinimidyl ester (Ax700-SE), a small molecule fluorophore that covalently binds free amines on cellular proteins and is excluded from live cells with an intact plasma membrane. These properties make SE dyes like Ax700 excellent reagents for marking dying cells whose membrane is no longer able to exclude these compounds. Note Ax700-SE differs from carboxy-fluorescein-SE (CFSE), which is able to cross an intact membrane.

For more information and a protocol for how we use Ax700-SE for testing viability in cytometry experiments, see Table 2 in Doxie & Irish *Curr Top Microbiol Immunol*. 2014 PMID: PMC4216808. We have added clarifying text and this reference to lines 470-472 of the Results under the section titled “Diversity Set Experiment”.

Lines 504-506: “Briefly, cells were stained for viability with 0.04 $\mu\text{g mL}^{-1}$ Alexa 700 SE (Ax700-SE), fixed with 1.6% paraformaldehyde, and permeabilized with 100% ice-cold methanol (for more information on the use of Ax700-SE for viability testing, see Table 2 in [Doxie & Irish *Curr Top Microbiol Immunol*. 2014 PMID: PMC4216808]).”

13. The authors write both “arcsine” and “arcsinh” (figures and text). Are these the same thing?

We appreciate this question from the reviewer. The arcsin function is the inverse of the sine function, and arcsinh is the inverse hyperbolic sine (arcsinh is the hyperbolic arcsin, a different mathematical operation).

To clarify this language, we have modified the text “hyperbolic arcsin scale” on lines 558-559 to instead say “inverse hyperbolic sine (arcsinh) scale” to minimize confusion. For more details on scaling using the inverse hyperbolic sine in cytometry analysis, see original use in the Supplement of PMID: PMC2919949, and recent implementations PMID: PMC3627543, PMID: PMC5750220, and PMID: PMC8370768.

Reviewer #2 (Remarks to the Author):

This article, titled "Connecting Chemical Structure to Single Cell Signaling Profiles" by Thirman et al., presents an approach called SAR-MAP (Structure-Activity Relationship using Multiplexed Activity Profiling) for high throughput ex vivo characterization of cellular activities in response to multiple chemical structures. A strength of this work is the use of fluorescent cell barcoding to measure multiple cellular readouts

simultaneously at the single-cell level (note that this method was published previously). The method is interesting and has the potential to be applied to study other cells and molecules.

We thank the reviewer for feedback on the strengths and potential future applications of this study.

1. More details are needed re the FCB procedure. The labeling steps are included in the method, but it is not entirely clear to this reviewer what cells are multiplexed. Each well on the plate is for a different compound at a different concentration, and then these wells were barcoded/mixed? Some type of schematic representation would go a long way for readers not familiar with this technique.

We thank the reviewer for this note. We recently published a protocol describing the FCB procedure in detail (see Schares et al., *Methods in Cell Biology* 2025 PMID: 40180452). To address the confusion around FCB within the manuscript, we have added additional text to the Methods section titled “Fluorescent Cell Barcoding (FCB) Assays” (lines 481-484). Additionally, we have added a new figure to the Supplementary Information (Supplementary Figure 1) depicting the version of the barcoding assay used here - including specific levels of Pacific Orange, Pacific Blue, and Alexa Fluor 750 - used to barcode each well of the experiment.

Lines 481-484: “Fluorescent cell barcoding (FCB) is a multiplexing approach where cells within a given well are covalently labeled with different discrete levels of more than one N-hydroxysuccinimidyl ester functionalized amine reactive fluorescent dyes. Using FCB, each well has a distinct fluorescent signature or barcode; this enables samples to be pooled for processing and running on the cytometer, thus reducing reagent consumption and increasing throughput.”

Supplementary Figure 1 – Fluorescent cell barcoding uses a combination of varying concentrations of Pacific Orange, Pacific Blue, and Alexa Fluor 750 to assign each well a unique combination of dyes. a) Rows of cells were fluorescently barcoded using 8 different concentrations of Pacific Blue. b) Columns of cells were fluorescently barcoded using 2 sets of 6 different concentrations of Pacific Orange. c) Alexa Fluor 750 was used at a single concentration as a dye uptake control. These three dyes are all used together on one plate to provide each well with a unique signature of Pacific Orange, Pacific Blue, and Alexa Fluor 750.

2. The heterogeneity within MV4;11 is intriguing but also raises questions. This is a cancer cell line which is clonal at least at the DNA genomics level. What is driving the variation in drug response? How do we know these are not just stochastic? It would be a lot more convincing if this is done using PDX-derived cells which often consist of multiple clones etc.

We appreciate this point from the reviewer and have added a paragraph to the Discussion (lines 369-381) addressing the point about variation in observed responses within populations of clonal cells and ways this could be further studied.

3. As only one cancer cell line and one control are studied, the validity of the biological discoveries in terms of the activity and selectivity of the prioritized compound may need further studies to confirm. I suggest the authors mention this in the discussion. Or they could consider mixing two different cell lines each with a differential drug response profile.

We appreciate this note from the reviewer. To address this concern, we have added new data from two cell types, H524 lung cancer cell line and healthy mouse embryonic neural stem cells (eNSCs). This doubles the number of cell types investigated as part of this study and validates that rocaglates are robustly bioactive in healthy cells and cancer cell lines (Supplementary Figure 3 and page 3 above). Additionally, we clarified which key results have been replicated and repeated (lines 414-416) and where further validation in follow-up studies will be valuable (lines 411-413, lines 418-420).

Lines 414-416: "The prominent result of cooccurring γ H2AX, p-4EBP1, and p-S6 S240/244 activity from the top two molecules identified - CMLD012390 and CMLD013342 - replicated in the two additional studies presented, thus providing encouraging evidence that the SAR-MAP platform is reliable and reproducible."

Lines 411-413: "Due to the inherent variability of cells, even within a clonal population, future studies should include replication of the Rocaglate Set screen on MV411, other leukemia cell lines, and primary leukemia samples to validate the contrasting bioactivity and signature profiles across structural subclass."

Lines 418-420: "Thus, future experiments should include a time course and testing on primary leukemia with cell surface markers to resolve the cellular mechanisms and subpopulations driving the MADD signaling profile."

5. PBMC includes a multitude of cell types and these experiments (I assume for safety/toxicity signals) are cursory and uninformative

We thank the reviewer for this note and recognize that PBMC includes multiple cell types, including lymphocytes like T cells, B cells, and NK cells and myeloid origin cells like monocytes. We also note that lymphocytes in healthy blood are low in side light scatter (Side Scatter) due to being small, non-complex cells, whereas myeloid origin cells like monocytes are higher in side light scatter due to complex or granular membrane and organelle structures. Thus, the y-axis of Side Scatter is traditionally a good surrogate for distinguishing myeloid cells (here, mostly monocytes) and lymphocytes (here, mostly T cells and a minority of B cells and NK cells).

We have added additional text to the Results section describing Figure 4c regarding the ability to distinguish these cell types within PBMC and to compare their responses to rocaglates (lines 239-251). Additionally, we added text to the Discussion describing the potential and limitations for future experiments to include cell surface protein detection in order to definitively separate specific subsets cells, such as memory T cells, as this is a possibility for multidimensional flow cytometry (lines 379-381, lines 418-420).

Lines 379-381: "Further multiplexed flow cytometry studies that include cell surface markers, single-cell RNA-sequencing, and validation on patient-derived xenografts might be performed to elucidate drivers of this variation in drug response."

Lines 418-420: "Thus, future experiments should include a time course and testing on primary leukemia with cell surface markers to resolve the cellular mechanisms and subpopulations driving the MADD signaling profile."

4. The EC50 calculation is not clear to this reviewer. Since similarity with the MADD score can be calculated for each cell, why is clustering (by DBSCAN) necessary and why only selected clusters are used? There is a steep change in MADD signaling between 0.1 to 0.5 μ M for the two RPs, resulting in EC50 of ~600nM. This seems to suggest that the effective concentration ranges of RPs to activate MADD are narrow. However, the concentration used for profiling was 10nM, which is much lower than the concentration needed to activate MADD in the two RPs. Can the authors explain this discrepancy?

We appreciate the reviewer's note regarding confusion with the EC50 calculation and have added clarifying text and new Figures in order to better make these points. To address the specific points: similarity with the

MADD score can be calculated for each cell using Velociraptor-Eye alone. DBSCAN was applied at the conclusion of the workflow as a way of grouping cells identified by Velociraptor-Eye based phenotype (here the response across phospho-proteins) for the EC50 calculation.

We appreciate the use of DBSCAN may have been confusing and have changed the approach here; the EC50 re-calculated based solely on Velociraptor-Eye score, as suggested by the reviewer (all cells with >75% similarity to the MADD signaling profile search were used). Data analyzed this way are included in Figure 6c, Figure 6d, and Supplementary Figure 9. The concentration used for the tests (10 μM) was roughly 17-fold above the EC50 (~0.6 μM); it is typical for such single cell assays to use a 10- to 20-fold higher than EC50 dose as a 'high dose' when seeking effects on cells and then to titrate to explore specific dose-response relationships for compounds (PMID: 18157122).

Figure 6 – Rocaglate pyrimidinones with mTOR activity during a DNA damage response possessed structural commonalities. a) t-SNE from Figure 2b of all compounds in MV411 divided based on rocaglate well of origin is shown for the three vehicle wells, etoposide, nocodazole, and a representative amidino rocaglate (ADR) (**CMLD013608**). The percentage of cells in the mTOR activity during a DNA damage response (MADD) signaling profile gate circled in dark blue is included at the bottom right of each plot. **b)** t-SNE from Figure 2b of all compounds in MV411 divided based on rocaglate well of origin is shown for the set of 9 RPs in order of decreasing percentage of cells with the MADD signaling

profile (t-SNEs for all individual rocaglates are shown in **Supplementary Figure 4a**). The percentage of cells with the MADD signaling profile is included at the bottom right of each plot. The chemical structure corresponding to each rocaglate is depicted on the right side of each t-SNE plot. **c)** Schematic depicting dose response experimental workflow and analysis. **d)** Dose-response titration curves depicting the abundance of MADD-like cells vs. $\log[\text{dose}(\text{nM})]$. A half-maximal activating concentration (EC₅₀) curve as fitted to the data and EC₅₀ values are shown in the middle of each plot. An EC₅₀ of 585 nM ($p < 0.001$) was generated for **CMLD012390** and EC₅₀ of 555 nM ($p = \text{n.s.}, \alpha = 0.05$) was generated for **CMLD013342**.

Reviewer #3 (Remarks to the Author):

Thirman et al. performed a diversity library screen by flow cytometry, identifying Rocaglates as a chemical class with specific activity on leukemic cells. Whilst the pharmacology on leukemia cells has been previously demonstrated, the authors go on to leverage multiparametric flow cytometry in order to characterize subclasses of molecules exhibiting differentiated pharmacology, and relate these back to structural features of the molecules.

Overall, certain claims are made around novelty that I'm unsure of. For example, Rocaglates anti-leukemic activity is well established (Callahan et al. - <https://ashpublications.org/blood/article/120/21/1338/89568/Rocaglamide-Selectively-Eradicates-Human-Leukemia>). Several other papers have used multi-parametric flow cytometry endpoints (e.g. see Ding et al. - <https://pubmed.ncbi.nlm.nih.gov/30807094/>; Tuijnberg et al - <https://pubmed.ncbi.nlm.nih.gov/31621069/>). The integration with multiparametric visualization is a slight advance here, however the data is used in a more qualitative way to describe and bin compound subclasses. Finally, claims about the utility of these approaches as they relate to drug discovery are a stretch, given that it's unclear to me why multiplexing the endpoints in the same assay is advantageous over simply running several, discrete, medium throughput assays. However to this final point, authors could perhaps discuss the importance of generating multiparametric data for natural product/phenotypic screening libraries early on in a drug discovery campaign versus sequential generation of this data. What was most interesting was the authors delineation of subclasses of Rocaglates, which were previously considered to being a single class. In particular, the MADD signature, represented in a small fraction of cells (but potentially indicative of a broader activity in the entire population) significantly highlights the benefits of integrating single-cell techniques into early SAR studies. This for me is the key advance in the paper, that could be elaborated and emphasized more in the conclusions.

We thank the reviewer for these notes. We have clarified why multiplexing endpoints for early drug discovery, particularly medicinal chemistry applications, is more advantageous than sequential single readout assays in our response below. We appreciate notes from the reviewer on novelty of the delineation of rocaglate subclass and the MADD signaling profile discovered using multi-dimensional, single-cell approaches; we have added additional text to the discussion emphasizing this point.

Major:

1) Some of the pharmacology terminology should be improved, to avoid confusion with more standard use. I would recommend replacing 'bioactivity' with 'pharmacodynamic effects', as it is referring to a specific phenotypic readout (gH2ax induction). 'Selectivity' tends to be used to refer to specific molecular interactions, and should be replaced with 'selective activity'. 'target activity' should be referred to as 'proximal target engagement'. This should be amended throughout the manuscript and figures.

We appreciate this point from the reviewer. In response to this feedback, we have replaced uses of the term "selectivity" with "selective activity", and "target activity" with "proximal target engagement" throughout the manuscript and in Supplementary Table 1. We have closely reviewed and updated use of the term "bioactivity" to ensure it is referring to a set of phenotypic readouts and used the term "pharmacodynamic

effect” when talking about things like a dose response for a specific compound and readout pair. Updated definitions are now present in the first paragraph of the Results (lines 57-75). Bioactivity is defined as, “the ability of a molecule to broadly elicit a biological response compared with a vehicle control”, whereas pharmacodynamic effect is defined as, “potency in proximal target engagement (i.e., eIF4A1 clamping for rocaglates)”.

2) Related to 1, some of the terminology is inconsistent throughout the manuscript. pg.6 paragraph 1 is a specific example of this: “Exceptional activity following treatment with a compound can be measured in different ways, including exceptional target activity or potency (i.e., clamping eIF4A1 at a given concentration, half maximal inhibitory concentration)”. This statement is simply referring to potency and efficacy parameters for a measure of proximal target engagement. The authors should take a careful look at the pharmacology terms throughout the paper to ensure consistency and clarity.

We thank the reviewer for this note. We have redefined the pharmacology terminology based on point 1 above in the first paragraph of the Results section (lines 57-75) and adapted the terminology used throughout the remainder of the manuscript to be consistent with these initial definitions.

3) The 5th paragraph in the introduction should be removed, and integrated with the first paragraph of the results section.

We thank the reviewer for this suggestion and have removed the 5th Introduction paragraph and incorporated its content within the first, third, and fourth paragraphs of the Results section, as discussed above.

4) Throughout the manuscript, data is all generated and interpreted at a single (high) concentration. Whilst this approach makes sense for selection of initial compounds, it is very challenging to interpret the rest of the manuscript without an understanding of the dose response. The way I might resolve this is to explore dose responses for some key compounds and major conclusions from the paper. For example – in figure 3A, it would be helpful to understand whether some of the less active compounds are simply a result of where they sit on their dose response curve versus more active compounds - i.e. is the SAR observed related to potency, efficacy or distinct pharmacology?

We appreciate this point from the reviewer and have updated the figures to highlight dose response results for three key compounds (CMLD013342, CMLD012390, and CMLD013608) and have added dose response data to the main text (lines 291-300) and figures (Figure 6, page 10 above). These results provide clear evidence that the SAR observed did not result from differences in potency for one mechanism, further supporting the idea that the SAR results from distinct pharmacology.

Lines 291-300: “A dose-response experiment was performed in MV411 as an assessment of the reproducibility of MADD signaling profile activation by the exceptional RPs. The top two exceptional RPs, CMLD012390 and CMLD013342 were chosen for testing alongside CMLD013608, a representative ADR, as a control (Figure 6c, Supplementary Figure 9a). Velociraptor-Eye (VR-Eye), a recently developed cell identification algorithm, was then used to identify the cells with the MADD signaling profile within the dose response dataset (Figure 6c, Supplementary Figure 9b) [PMCID: PMC11092669]. The resulting cells with greater than 75% similarity to the MADD signaling profile MEM label were used to calculate the half-maximal activating concentration (EC50) for each of the two RPs (Figure 6c). CMLD012390 and CMLD013342 had similar EC50s of 585nM and 555nM, respectively (Figure 6d). Thus, these two RPs demonstrated a reproducible, dose-dependent induction of this signature profile.”

5) It would greatly add to the paper to perform basic cell killing assays (beyond simply gH2AX or cleaved caspase), and to show differential pharmacology of the MADD compounds versus non-MADD profile compounds. For example, demonstrating a differential between proximal target engagement potency and

cell killing that correlates to the MADD characteristics would show biological significance of this finding. As it stands the results are fine but somewhat descriptive, whereas linking the subclasses of compounds back to a biologically meaningful phenotype would be a powerful demonstration of approach taken.

We thank the reviewer for this suggestion. To quantify basic cell killing and demonstrate the distinction between basic cell killing and activation of the MADD signaling profile, we added panels b and d to Figure 3 (page 5, above); these depict cell killing by quantifying fold change in cell number (event count, compared to lowest cell counts in black to the highest cell counts in bright yellow). We also note that the gating strategy filters out dead cells (cells lacking an intact plasma membrane) during analysis (lines 190-193). Combined with the barcoding approach, the relatively short time frame of the experiment (16 hours), and the use of multiple per cell readouts of proliferation, cell cycle, and cell death mechanisms, the relative decrease in cell number in treated wells here serves as a measure of cell death. Minimal variation in this measure can be observed across the vehicle replicates.

Lines 190-193: “Additionally, the fold change in cell count for treated wells as compared to vehicle cell count was calculated (i.e. $\frac{\text{cell count}_{\text{treated}}}{\text{cell count}_{\text{vehicle}}-1}$). Given that this dataset was gated for live, intact, single cells as part of pre-processing, cell count acts as a measure of basic cell killing.”

6) The advantages of multiplexing assays (e.g. γ H2ax, phosphor endpoints), versus more traditional and higher throughput approaches such as immunofluorescence should at a minimum be discussed. Typically in a drug discovery setting, running multiple assays in medium throughput is not rate limiting, and its unclear whether the sensitivity of the flow cytometry assay is comparable to other more established methods such as γ H2AX imaging. This being said, generating multiparametric data on initial compounds rather than sequentially can certainly have advantages to choose starting matter or accelerate med chem efforts (vs finding out later into a med chem campaign).

We appreciate this point from the reviewer. We have added text to the Discussion section (lines 401-408) pertaining to the application of SAR-MAP for deconvolving the impact of the 4'-methoxy substituent; here, we describe how multiparametric data not only enables small chemical changes to be more efficiently discerned but reveals the signal nodes simultaneously perturbed. Additionally, we now discuss the sensitivity of flow cytometry assays for detection of features like γ H2AX and have added references to papers documenting the rigor and reproducibility of this assay, limitations, and advantages, as compared to traditional approaches, including microscopy (lines 353-356).

Lines 401-408: “Follow-up SAR-MAP confirmed that a shift from a 4'-methoxy substituent to a 4'-bromine led to decreased p-4EBP1 and p-S6 S240/244 activity and c-CAS3 activation. This marked shift in signaling profile detected via phospho-flow highlights an advantage of multiparameter assays versus performing multiple sequential single readout assays for accelerating medicinal chemistry efforts. Having a larger biological resolution not only enables the impact of small chemical changes to be more efficiently discerned but reveals mechanistic information about the signal nodes simultaneously perturbed. Here, this includes the added knowledge that compounds with a 4'-methoxy substituent activate a DNA damage response without initiating apoptosis or suppressing mTOR pathway activity.”

Lines 353-356: “Though analysis of γ H2AX via microscopy-based approaches is considered to be the most sensitive method of detection, it is time consuming and operator-dependent [PMID: 29685191]. Flow cytometry has been validated as a reliable and sensitive measure of γ H2AX and is more efficient, automated, and can be detected alongside multiple additional readouts [PMID: 27060560].”

Minor:

7) More information should be provided on the ‘diversity set’ of compounds used for initial profiling. E.g. was this a commercial or proprietary compound library, were compounds all natural product derived, are

their previous publications characterizing the library? This can help to better contextualize the hits observed in figure 1.

We appreciate this note from the reviewer on the need for additional information on the Diversity Set to better contextualize Figure 1. We added text to the Methods section titled “Curation of Compound Libraries” (lines 464-479) providing additional information on the compounds in the Diversity Set.

8) A representative of the amidino-rocaglate structural subclass should also be shown in supplementary figure 1 to better enable comparison of structure

We appreciate this suggestion from the reviewer. To depict a representative of each rocaglate subclass, CMLD012824, an exemplar ADR, was added to the figure to enable better comparison of structure. This is now Supplementary Figure 2.

Supplementary Figure 2 – RocA, CMLD012824, and aglaroxin C were exemplar rocaglates for their respective subclasses. a) Chemical structure for rocaglamide (RocA), a regular rocaglate (RR). R group is colored red based on classification as RR. b) Chemical structure for CMLD012824, an amidino rocaglate (ADR). Ring fusion is colored orange based on classification as ADR. c) Chemical structure for aglaroxin C, a rocaglate pyrimidinone. Ring fusion is colored in green based on classification as RP.

9) The introduction should also include mention/citation of the mechanism of action of rocaglates that underpins their selective activity in leukemia cells – e.g. if the differential effects on leukemia vs normal is due to the high growth/proliferation rates of leukemic cells.

We thank the reviewer for this point. We added text to the Introduction (lines 85-88) clarifying potential mechanisms underlying the leukemia selectivity of rocaglates.

Lines 85-88: “Protein synthesis is an attractive cancer target as it is tightly regulated in normal cells but often highly dysregulated in cancer cells, leading to uncontrolled growth and survival²⁸. As such, rocaglates have been cited for their ability to selectively induce cancer cell death while minimally disrupting healthy cells, specifically in leukemia [PMID: 17565740, PMCID: PMC4148474]”

10) Pg. 6 paragraph 1. “Exceptional activity following treatment with a compound can be measured in different ways, including exceptional target activity or potency (i.e., clamping eIF4A1 at a given concentration, half maximal inhibitory concentration..” It is very challenging to make claims about exceptional activity based on activity ‘at a given concentration’. Rather, the activity should be based on potency or efficacy of a compound. This should be clarified in the text

We appreciate this note from the reviewer and have removed the text defining exceptional activity as eIF4A1 clamping at a given concentration. Additionally, we believe that the changes to the pharmacology terminology described in points 1 and 2 above will help clarify definitions of activity.

11) Figure 4C - is it understood why the side scatter parameter is so variable between the examples shown? Was SSc correlated with any other parameters across compounds?

We thank the reviewer for this note about side light scatter (Side Scatter or SSC). Additional text discussing

how SSC is a surrogate of cellular features for PBMC and what variation in SSC means for MV411 have been added to the Results section for Figure 4c (lines 239-251).

12) Pg.9 gH2AX induction can be a function of compound potency, efficacy and time. In addition, it is not clear why a 15% cut off is used to determine if compounds trigger DNA damage. Overall this is a fairly qualitative way to evaluate compounds. The gating strategy (Fig3C) is also questionable, in that very small shifts in MFI are having a large impact on the % positive due to the tight gating. An orthogonal assay (e.g. gH2AX immunofluorescence assay) would build confidence in the authors conclusions.

We thank the reviewer for this note about γ H2AX induction. In response to this feedback, we have adapted the cutoff to be based on a Wilcoxon rank sum test that was conducted utilizing the % γ H2AX+ values in MV411 and PBMC for the 37 rocaglates. An alternative hypothesis that the true median % γ H2AX+ was greater than the median vehicle % γ H2AX+ was utilized. The confidence interval generated by this hypothesis test, 13.3%, was taken as the significance threshold. We have adapted the text (lines 217-223) and Figure 4a to reflect this change in threshold.

We also appreciate the points about the gating strategy; to add additional confidence that the percent in gate values accurately reflect the shifts in MFI, we have added the raw MFI values to Figure 4c. Calculating percent in gate to evaluate and compare marker expression is a commonly used metric in analysis of cytometry data (see Earl et al. *Nature Communications* 2018, PMID: PMC5750220, Balsamo et al. *J Biol Chem.* 2022, PMID: PMC9424577).

Lines 217-223: “To distinguish compounds that triggered significant DNA damage in MV411 and PBMC a threshold of vehicle median + 3 x vehicle IQR was first considered; this is the cutoff that was utilized in Figure 1a and Figure 2d. However, at this cutoff (7.0% γ H2AX+), all except 6 rocaglates triggered significant DNA damage in MV411; this is another indication assay quality and exceptional rocaglate bioactivity. To develop a more stringent threshold, a Wilcoxon rank sum test was conducted utilizing the % γ H2AX+ values in MV411 and PBMC for the 37 rocaglates; an alternative hypothesis that the true median % γ H2AX+ was greater than the median vehicle % γ H2AX+ was utilized. The confidence interval generated by this hypothesis test, 13.3%, was used as the significance threshold. “

Figure 4 – MV411 specific activity was observed in the rocaglate pyrimidinone subfamily and some regular rocaglates. **a**) Bar plot of percentage of γ H2AX positive (% γ H2AX+) cells for each compound grouped and colored according to rocaglate subclass listed on the right side of the plot. Bars on the left side of the vertical black line correspond to the compound response in peripheral blood mononuclear cells (PBMC) and the right side corresponds to the compound response in MV411. The dotted vertical pink and blue lines correspond to a significance threshold of 13.3% γ H2AX+ cells in PBMC and MV411, respectively. Compounds that cross the threshold have darkened colored bars. The compound name (or vehicle) is listed on the far left side of plot colored according to the following system: blue = increases % γ H2AX+ cells past threshold in MV411 and not in PBMC, pink = increases % γ H2AX+ cells past threshold in PBMC and not in MV411, light grey = does not increase % γ H2AX+ cells past threshold in either cell type, dark grey = increases % γ H2AX+ cells past threshold in both cell types. Arrows are displayed to the left of compounds that are shown in **Figure 4c**. **b**) Scatter plot of % γ H2AX+ cells in MV411 on the x-axis and the log2 fold ratio of % γ H2AX+ cells in MV411 to % γ H2AX+ cells in PBMC on the y-axis. Each dot corresponds to a compound colored according to rocaglate structural subclass. Compounds that will be shown in **Figure 4c** are circled and labeled. **c**) Contour plots of Side Scatter vs. γ H2AX for Vehicle 3, **CMLD010508**, **CMLD012600**, and **CMLD013342**, respectively from left to right in PBMC (top) and MV411 (bottom). The contour plot was drawn with 10% of cells per contour and outliers starting at 10%. The blue line indicates the expert drawn γ H2AX+ gate. The % γ H2AX+ cells within the gate and the median fluorescence intensity are written in blue in the lower right corner. Corresponding structures for each compound are depicted below the compound name.

13) The dose response data in Supplementary Figure 7 is very nicely supporting of the qualitatively distinct SAR, and could be incorporated into the main text, as part of Figure 5.

We appreciate this suggestion from the reviewer about incorporating the dose response data from Supplementary Figure 7 into the main text and have made these changes [Results text (lines 291-300) and

dose response data added to main Figure 6, (page 10 above)].

Lines 291-300: “A dose-response experiment was performed in MV411 as an assessment of the reproducibility of MADD signaling profile activation by the exceptional RPs. The top two exceptional RPs, CMLD012390 and CMLD013342 were chosen for testing alongside CMLD013608, a representative ADR, as a control (Figure 6c, Supplementary Figure 9a). Velociraptor-Eye (VR-Eye), a recently developed cell identification algorithm, was then used to identify the cells with the MADD signaling profile within the dose response dataset (Figure 6c, Supplementary Figure 9b)³⁷. The resulting cells with greater than 75% similarity to the MADD signaling profile MEM label were used to calculate the half-maximal activating concentration (EC50) for each of the two RPs (Figure 6c). CMLD012390 and CMLD013342 had similar EC50s of 585nM and 555nM, respectively (Figure 6d). Thus, these two RPs demonstrated a reproducible, dose-dependent induction of this signature profile.”

14) Authors could consider citing the following review articles, or integrating some of their cited references: Ullas et al. (<https://pubmed.ncbi.nlm.nih.gov/38612661/>), which has a section on some of the ways flow cytometry has been applied to early phenotypic screening; Vincent et al. (<https://pubmed.ncbi.nlm.nih.gov/35637317/>), which highlights the limitations of single-parameter phenotypic screening, and includes discussion of applying multiparametric endpoints.

We thank the reviewer for suggesting additional articles to cite. We have incorporated content from both references, in addition to some of their cited references, into the first paragraph of the Introduction (lines 37-42).